

# Text-mining forma mentis networks reconstruct public perception of the STEM gender gap in social media

Massimo Stella

Complex Science Consulting, Lecce, Italy

## ABSTRACT

Mindset reconstruction maps how individuals structure and perceive knowledge, a map unfolded here by investigating language and its cognitive reflection in the human mind, i.e., the mental lexicon. *Textual forma mentis networks* (TFMN) are glass boxes introduced for extracting and understanding mindsets' structure (in Latin *forma mentis*) from textual data. Combining network science, psycholinguistics and Big Data, TFMNs successfully identified relevant concepts in benchmark texts, without supervision. Once validated, TFMNs were applied to the case study of distorted mindsets about the gender gap in science. Focusing on social media, this work analysed 10,000 tweets mostly representing individuals' opinions at the beginning of posts. "Gender" and "gap" elicited a mostly positive, trustful and joyous perception, with semantic associates that: celebrated successful female scientists, related gender gap to wage differences, and hoped for a future resolution. The perception of "woman" highlighted jargon of sexual harassment and stereotype threat (a form of implicit cognitive bias) about women in science "sacrificing personal skills for success". The semantic frame of "man" highlighted awareness of the myth of male superiority in science. No anger was detected around "person", suggesting that tweets got less tense around genderless terms. No stereotypical perception of "scientist" was identified online, differently from real-world surveys. This analysis thus identified that Twitter discourse mostly starting conversations promoted a majorly stereotype-free, positive/trustful perception of gender disparity, aimed at closing the gap. Hence, future monitoring against discriminating language should focus on other parts of conversations like users' replies. TFMNs enable new ways for monitoring collective online mindsets, offering data-informed ground for policy making.

# INTRODUCTION

Perception is in the mind of the beholder. Every experience contributes to building a memory or mental reconstruction of the outer world which, in turn, deeply impacts future behaviour (*Malt, Wolff & Wolff, 2010*; *Aitchison, 2012*; *Beasley & Mason, 2015*; *Seli et al., 2019*; *Li, Baucom & Georgiou, 2020*). Negative perceptions can inhibit efficient learning (*Chavatzia, 2017*) or drastically alter information processing (*Shapiro & Williams,*

Corresponding author
Massimo Stella,
massimo.stella@inbox.com

*2012*), while positive perceptions can contribute to the acceptance and establishment of social norms (*Malt, Wolff & Wolff, 2010*; *Waterloo et al., 2018*; *Welles & González-Bailón, 2020*). In this way, reconstructing and understanding the cognitive perceptions of social groups is key to achieving insights about human behaviour and social patterns.

Discovering the perception of a given audience is a challenge that can be broken mainly in two parts: (i) acquiring cognitive data, reflecting how the public perceives a certain phenomenon, and (ii) processing such data in an efficient way, suitable for the extraction of new knowledge (*De Arruda et al., 2019*). Data can be gathered by mining social media (*Stella, Ferrara & De Domenico, 2018*; *Bovet, Morone & Makse, 2018*), which represent an invaluable source of information about the users' experiences and perceptions of specific topics (*Welles & González-Bailón, 2020*). Recently, social media, Twitter in particular (*Jansen et al., 2009*), have been increasingly analysed by the scientific community in order to detect complex phenomena such as the emotional dynamics of voting events (*Stella, Ferrara & De Domenico, 2018*; *Bovet, Morone & Makse, 2018*), the promotion of self-branding and journalistic content also through social bots (*Varol & Uluturk, 2020*), the spread of disinformation (*Pierri, Artoni & Ceri, 2020*) and the fostering of online hate dissemination (*Waqas et al., 2019*).

## Towards a cognitive approach to information processing

Although Twitter messages provide several types of information, such as visual cues (e.g., pictures and videos) or multi-language cues (e.g., emojis and hashtags), their nature is mainly textual (*Welles & González-Bailón, 2020*; *Jansen et al., 2009*). In this way, the problem of quantifying how an audience perceives a given topic can be related to stance detection from their tweets (*Mohammad, 2016*). Approaching stance detection through human coding becomes quickly intractable when faced with the large volumes of messages exchanged daily on the Twittersphere. This limitation leads to the above need for developing and adopting efficient techniques for knowledge extraction from massive amounts of text as exchanged on social media. Given the extraordinary possibility for humans to communicate their mental constructs through language (*Aitchison, 2012*), a key way of detecting perceptions is through communication. Both in computer science and linguistics, the problem of detecting positive or negative perceptions from language is known as *stance detection* (*Mohammad, 2016*). Rather than focusing on language in itself, this work shifts the attention to the cognitive reflection of language in the human mind, in the so-called *mental lexicon* (*Malt, Wolff & Wolff, 2010*; *Aitchison, 2012*; *Dóczi, 2019*; *De Deyne et al., 2019*). Such a lexicon includes semantic memory, a well-studied repository of conceptual meanings and word features (*Kenett et al., 2017*; *Dóczi, 2019*), and also other memory supports storing syntactic/phonological/orthographic and even affective knowledge, together with other aspects of language (*Dóczi, 2019*), whereas syntax, phonology and orthography express word-level conceptual knowledge (*Aitchison, 2012*), affective knowledge links concepts to the emotions they elicit (*Malt, Wolff & Wolff, 2010*); therefore, it represents an important component of any stance/perception.

Harnessing the complex cognitive structure of the mental lexicon means accessing how conceptual knowledge is structured and emotionally perceived by individuals as the

outcome of their previous experiences and current attitudes (*Aitchison, 2012*; *Stella et al., 2019*; *Stella, 2020*). In other words, accessing the conceptual and affective representation of knowledge in the mental lexicon means reading minds(ets), with strong repercussions for information processing. Discovering how an audience perceives a given topic, i.e., performing mindset reconstruction, can provide crucial knowledge for understanding and intervening upon specific trends (*Welles & González-Bailón, 2020*; *Amancio, Oliveira Jr & Costa, 2012*; *Stella et al., 2019*). An example is represented by the finding that mindsets vehiculating positive emotions reach larger audiences on social media whereas negative emotional content can spread at faster rates (*Ferrara & Yang, 2015*). Another example for the relevance of mindset reconstruction is uncovering and acting upon traces of science anxiety in student populations in order to improve their learning experiences (*Stella, 2020*; *Stella & Zaytseva, 2020*) or detecting sexual harassment through large-scale web surveys (*Karami et al., 2020*).

A key area where mindset reconstruction is particularly promising is understanding the sources and dynamics of the *gender gap in science* (*Hogue & Lord, 2007*; *Moss-Racusin et al., 2012*).

As outlined by recent Big Data Analytics studies considering decades of scientific careers (*Huang et al., 2020*; *Odic & Wojcik, 2019*; *Chavatzia, 2017*), gender disparities in science cannot be explained by intrinsic differences in attitudes to science between genders but rather have to be traced in the establishment of implicit gender biases promoted by news media and by social media representations of science (*Shapiro & Williams, 2012*; *Moss-Racusin et al., 2012*; *Madsen & Andrade, 2018*; *Steinke, 2017*). In this way, the reconstruction of mindsets vehiculated by information systems becomes a key point for understanding, acting upon and closing the gender gap in science. Given the recency of the above mentioned Big Data Analytics studies and the methodological issues in reconstructing mindsets with black-box machine learning techniques (often neglecting contextual information; *Nasar, Jaffry & Malik, 2019*), reconstructing online perceptions of the gender gap in social media is a challenge not fully explored yet.

## Research aim

This work introduces textual forma mentis networks (TFMNs) as quantitative tools for reconstructing the mindset of online users engaging in social discourse with the research aim of investigating in detail the mindset emerging from user-generated content about the gender gap in science. In other words, this work uses textual forma mentis networks with the aim of reconstructing how online users discussed and perceived messages revolving around the STEM gender gap. In order to mirror the general perception of this gap, a window without special events about women in science was chosen. TFMNs quantified the general stance from 10,000 tweets publicly available on Twitter, produced between October 8 2019 and October 22 2019, containing the words or hashtags "science" or "stem" and "women" or "gendergap". These tweets encapsulated information about how the online authors perceived women in science, with relevant impact for education research and the computational social sciences.

Before providing additional details about the adopted methodology, it is necessary to compare TFMNs against past approaches, also relating forma mentis networks within the relevant literature about the gender gap in science.

## Literature review on relevant past approaches

It has to be underlined that only recently automatic text-mining studies started investigating gender gap in online discourses (*Teso et al., 2018*; *Chavatzia, 2017*). However, these approaches mainly focused on detecting differences in language use *among* different genders. Differently from other information processing investigations aiming at identifying emotions on social media in relation to phenomena like hateful speeches (*Waqas et al., 2019*) or disinformation spreading (*Pierri, Artoni & Ceri, 2020*), gender-focused investigations of social media did not explore large-scale mappings of the online perception of the gender gap in science as embedded in the messages exchanged between social users of any gender. Within an information management setting, the work closest to an investigation of the overall mindset about gender biases was the study by Karami and colleagues (*Karami et al., 2020*). The authors investigated online self-reports of sexual harassment experiences and through a topic analysis they highlighted evidence for sexual harassment in academia mainly targeting women and involving coercion, gender discrimination and retaliation. Building upon the knowledge extraction approach of *Karami et al. (2020)*, this study shifts its attention from explicit sexual harassment to the larger topic of gender biases in science, which includes harassment itself but also implicit biases (*Shapiro & Williams, 2012*), gender pay gaps (*Courey & Heywood, 2018*) and stereotypical perceptions about leadership (*Pennington et al., 2016*; *Ely, Ibarra & Kolb, 2011*). Furthermore, rather than focusing on self-reports, this study aims at tackling a different information system, namely Twitter, where social users can engage in social discourse and reach large audiences (*Welles & González-Bailón, 2020*). This reconstruction of social media attitudes about the gender gap in science is the main gap that the current work aims to fulfil by using a network-powered approach, which has several similarities and novelties in comparison to past network frameworks.

Within the literature of network-powered analyses of large volumes of text (*De Arruda et al., 2019*; *Koponen & Nousiainen, 2019*), previous works successfully used word co-occurrences in text (e.g., "like" and "stem" occurring one after the other in text, cf. *Cancho & Solé, 2001*) in order to characterise language content through average statistical markers (*Amancio, Oliveira Jr & Costa, 2012*; *Amancio, 2015*) or time-evolving dynamics (*Akimushkin, Amancio & Oliveira Jr, 2017*). These networks were considerably powerful at a global level and very successful in tasks like author identification (*Amancio, 2015*; *Akimushkin, Amancio & Oliveira Jr, 2017*; *De Arruda et al., 2019*). However, the validity of co-occurrence networks as representations of the mental lexicon at the microscopic level of individual conceptual associations has been recently reconsidered (*Ninio, 2014*; *Rizvi, 2018*). In fact, co-occurrences can lead to spurious conceptual links, whose influence vanishes in global statistical approaches and that do not represent syntactic similarities in text (*Ninio, 2014*). In order to overcome this limitation and achieve a more faithful microscopic representation of the mental lexicon, TFMNs

directly harness the full ensemble of syntactic associations of a sentence (e.g., "like" being a verb referring to the object "stem", cf. *Cancho, Solé & Köhler, 2004*) and enrich them by considering also semantic overlap between words (e.g., "appreciate" and "like" being synonyms across different sentences, cfr. *Miller, 1998*; *Amancio, Oliveira Jr & Costa, 2012*).

Textual forma mentis networks automatically extract these conceptual associations from text without requiring human supervision and are therefore suitable for processing large volumes of text. The resulting network structure is informative of the cognitive layout of conceptual associations emerging from a given textual corpus and hence represents how text authors organised, structured and associated microscopically their knowledge around topics and concepts. This makes TFMNs "glass boxes" (*Nasar, Jaffry & Malik, 2019*), where the knowledge structure of a certain stance can be accessed and directly read, differently from previous "black box" machine learning approaches which accurately reproduced the positive or negative nature of a stance without providing information on its semantic content (*Mohammad, 2016*; *Teso et al., 2018*; *Rudkowsky et al., 2018*; *Nasar, Jaffry & Malik, 2019*). In addition to conceptual associations, TFMNs are endowed also with sentiment labels, indicating the sentiment (*Warriner, Kuperman & Brysbaert, 2013*) and the basic emotions (*Ekman & Davidson, 1994*; *Mohammad & Turney, 2013*) elicited by a given concept in a population of individuals involved in behavioural studies. Sentiment scores of positive/negative affect are also called word valence in psycholinguistics (*Warriner, Kuperman & Brysbaert, 2013*; *Recchia & Louwerse, 2015*) and represent how positively or negatively a given concept was perceived in a behavioural study. Word valence and emotional profiles add more emotional contextual information about the stance reconstructed by conceptual associations (*Stella, 2020*). Although large-scale datasets about sentiment and emotions have only been recently made available to the scientific community by cognitive studies (*Warriner, Kuperman & Brysbaert, 2013*), they have quickly become predominant in predicting a wide variety of human behaviour (*Li, Baucom & Georgiou, 2020*) and information processing patterns such as consensus formation in social networks (*Konstantinidis, Papadopoulos & Kompatsiaris, 2017*) or information sharing on microblogs (*Ferrara & Yang, 2015*). The combination of network patterns and sentiment data is important, as considering only the frequency of sentiment-labelled content in short texts has been reported to lack interpretative contextual power for estimating how people really feel about a given topic (*Beasley & Mason, 2015*).

## Different types of forma mentis networks

As outlined above, text-based forma mentis networks represent multiplex lexical networks (*Stella et al., 2018*) of concepts interconnected through syntactic and semantic associations and enriched with sentiment and emotional labels. Previous works showed how lexical networks of concepts including multiple layers of associations (i.e., "multiplex") were better than single-layer complex networks at predicting a variety of cognitive processes involved in information acquisition (*Stella et al., 2018*; *Stella, 2019*) and search (*Castro & Stella, 2019*; *Siew et al., 2019*). However, these approaches did not explore the influence that emotions and sentiment can have over information processing (*Warriner, Kuperman & Brysbaert, 2013*; *Dóczi, 2019*; *Rudkowsky et al., 2018*; *Li, Baucom & Georgiou, 2020*). The

introduction of behavioural forma mentis networks (BFMNs) (*Stella et al., 2019*; *Stella & Zaytseva, 2020*) tackled such a gap by combining conceptual associations with sentiment scores and providing access to knowledge structure and positive/negative perceptions of concepts in a given mindset. However, mindset reconstruction of BFMNs comes at the cost of involving individuals in a cognitive experiment (cf. *Stella et al., 2019*). This limitation translates into the possibility of reconstructing only the mindsets of those who participated in the experiment, preventing the monitoring of remote systems like social media. Textual forma mentis networks do not require behavioural experiments but can be built starting from any written text, instead. As a consequence, TFMNs are suitable for the investigation of speech over online platforms, whereas BFMNs are unsuitable for such task. However, as outlined above, both TFMNs and BFMNs exploit the theoretical framework of mental lexicon representations (*Dóczi, 2019*) and language processing in cognitive network science (*Siew et al., 2019*). Relying on these foundations, TFMNs are applied here for reconstructing the mindset of online discourse about the gender gap in science, as motivated in the next section.

## Recent literature identifies a link between the gender gap and distorted mindsets

Overwhelming evidence indicates that the gender gap in science is a complex phenomenon deeply affecting society, economics and science advancement (*Ely, Ibarra & Kolb, 2011*; *Madsen & Andrade, 2018*; *Hogue & Lord, 2007*; *Pietri et al., 2018*). The gender gap in the scientific, technological, engineering and mathematical (STEM) disciplines is a disparity of how different genders enter in and progress through a career in science (*Shapiro & Williams, 2012*). Between 2014 and 2016, UNESCO estimated only around 30% of all female students in higher education enrolled in STEM-related fields of study at University level (*Chavatzia, 2017*). This gender gap in STEM education and participation is almost absent at the level of primary education but then becomes particularly apparent during upper secondary education, coincidentally with subject selection, and it gets worse at higher levels of education (*Ely, Ibarra & Kolb, 2011*; *Chavatzia, 2017*; *Hogue & Lord, 2007*).

Globally, only 28% of all the world's researchers are women. This disparity is deeply embedded in educational systems, in particular in terms of attitudes and perceptions towards science (*Shapiro & Williams, 2012*; *Hogue & Lord, 2007*; *Huang et al., 2020*). The PISA 2015 report *Excellence in Education* reported that across 35 OECD countries, only 22% of 15-years old girls intend to pursue a career in STEM, less than half the proportion (48%) of STEM driven 15-years old boys (cf. *Chavatzia, 2017*). In the last few years, a great attention has been devoted to explaining such gap by using either external, e.g., system embedded discrimination (*Chavatzia, 2017*; *Ely, Ibarra & Kolb, 2011*), or internal factors, e.g., implicit stereotypes (*Shapiro & Williams, 2012*; *Pietri et al., 2018*). Studies trying to explain gender gap through gender-based learning achievements reported a complex landscape, with boys and girls being more or less proficient than each other according to the task being measured (cf. *Chavatzia, 2017*). This complex mosaic of multiple findings makes it complicated to state that any gender is more or less advanced in any given STEM subject: the gender gap has to be rooted in more subtle forms of discrimination, stereotypes

and perceptions (*Moss-Racusin et al., 2012*; *Lane, Goh & Driver-Linn, 2012*). Despite this fragmentation, the data unanimously indicate that women are paid less and leave STEM careers way more than their male colleagues either at the University level or at subsequent stages of professional growth in science (*Courey & Heywood, 2018*).

Even professional impact in STEM is influenced by a strong gender gap. The recent longitudinal study by Huang and colleagues (*Huang et al., 2020*) considered professional impact of women in STEM through bibliographic Big Data spanning over 60 years of publications. The authors reported evidence for gender differences in the cumulative productivity of research output but not in the annual rate of publication or career-wise impact. Through a quantitative, longitudinal analysis, Huang and colleagues related such gender gap to dropout rates and differences in length of publishing careers between men and women. The authors could not explain such disproportion only in terms of intrinsic gender-based proficiency in STEM, thus providing additional large-scale evidence for the presence of strong contextual factors affecting everyone's experience of the STEM gender gap. Analogous differences through Big Data approaches were recently found also by Odic and Wojcik in psychology, a field where three in four students are women whereas three in four academic professionals are male (*Odic & Wojcik, 2019*).

## The need to expose distorted mindsets about the gender gap in information systems

The predominance of strong academic biases in contrast with educational patterns suggests the presence of hidden roots to the STEM gender gap, as indicated by several independent studies on the topic (*Pennington et al., 2016*; *Shapiro & Williams, 2012*; *Pietri et al., 2018*). Hence, understanding the overall experience and subsequent perception of this gender gap in a large audience can provide key elements for better detecting the presence of potentially subtle yet strong *gender-based stereotypes*, as perceived and communicated by the main actors of STEM. Given that most of these stereotypes can act at a subconscious level and take place without the explicit awareness of those perpetrating them (*Pennington et al., 2016*; *Madsen & Andrade, 2018*; *Lane, Goh & Driver-Linn, 2012*), detecting the presence of such distorted perceptions remains a difficult challenge.

Monitoring and detecting the diffusion of stereotypical endorsements in social media represents an important way of better understanding and countering gender discrimination in science (*Karami et al., 2020*), especially in the current society dominated by virtual social media (*Welles & González-Bailón, 2020*; *Jansen et al., 2009*).

## Manuscript organisation

This manuscript is organised in several subsections. The "Methods" section contains quantitative details about the implemented methodology and analysed data. The "Results" section is split in two subsections. The first part is a quantitative benchmark, reporting on the effectiveness for TFMNs in finding semantically relevant concepts and topic features in short texts annotated by authors. The second part explores the reconstructed mindset of online users towards the gender gap by considering the stance towards key domains and aspects of the gender gap, e.g., concepts like "woman", "man", "person", "scientist", and

related aspects, e.g., "gender", "gap" and "stem". The detected conceptual associations and emotional patterns are investigated and related with previous studies within the "Discussion" section.

## METHODS

This section introduces the following elements: (i) the dataset about the online perception of the gender gap in STEM as retrieved from Twitter, (ii) the methodology behind the construction of text-based forma mentis networks, and (iii) the cognitive data used for detecting conceptual overlap in meaning between words and their valence. This section also reports on the benchmark data used for testing TFMNs, namely the text "Complexity Explained" (https://complexityexplained.github.io/-Last Accessed:16/03/2020) and the behavioural forma mentis network of international STEM researchers analysed in (*Stella et al., 2019*; *Stella, 2020*).

### Twitter dataset

The main dataset used in this investigation was a collection of 10,384 tweets publicly available on Twitter and produced between October 8 2019 and October 22 2019. Tweets were gathered through the *ServiceConnect[]* function for Twitter crawling implemented in Mathematica 11. Crawling was performed in accordance with Twitter's policies via the account of Complex Science Consulting, which received authorisation by Twitter for research-focused text mining. Tweets were included in the dataset if they contained either the hashtags (or the words) "#science" or "#stem" and "#women" or "#gendergap". This means that the combinations required for the twitter query were: (i) "science" and "women", or (ii) "science" and "gendergap", or (iii) "stem" and "women", or (iv) stem and "gendergap". The query service implemented in Mathematica works analogously to the search service implemented on Twitter, which can search tweets both at the start of conversations or in middle of them as long as they contain the above hashtags or words. Searching for either hashtags of words and including only two requirements for query made the search itself flexible in identifying a variety of tweets over such a specific topic, in a time window outside of specific events focusing over the gender gap.

Users' replies and mentions containing the same keywords as above were included in the analysis. However, threaded tweets or users' replies and mentions not including the above keywords were not added to the dataset. This choice made the dataset more focused over individuals' original statements about the gender gap, giving more voice to those users starting a conversation (with the above considered keywords) in order to express their points of view. This focus was motivated by the cognitive grounding of TFMNs, which combines linguistic knowledge and emotional perceptions as directly coming from people's mindsets/mental lexica, without strong influences from other people's behaviours or stances that might be present in users' replies (*Tagg, 2015*).

Twitter's policy prevents the redistribution of these tweets outside of the Twitter platform. Nonetheless, for the sake of scientific reproducibility, the IDs of these tweets were attached to this manuscript as Supplementary Information.

Each tweet contained a short text. Pictures and emoticons were discarded. Hashtag characters were removed. Tweets with less than three words were not included in the analysis but constituted less than 1% of the whole dataset.

## Network construction

Text-based forma mentis networks were built in three different stages:

1. Extraction of syntactic relationships between concepts/words from a sentence;
2. Addition of semantic relationships (synonyms) between concepts as indicated by an external dataset;
3. Addition of valence labels ("positive", "negative", "neutral") to each single concept as indicated by another external dataset.

The above three steps were repeated for all the sentences in a given tweet. At the end, all connections between the extracted and labelled concepts formed a multiplex lexical network (*Stella et al., 2018*) where nodes represented words/concepts and were connected across multiple linguistic layers, namely: (i) a semantic layer indicating meaning overlap between words (e.g., "famous" and "notable" sharing the same meaning), and (ii) a syntactic layer indicating dependencies in meanings as encapsulated within a sentence. In general, syntactic structure defines how entities (e.g., nouns) are specified in a given sentence through verbs, determiners, prepositions, adjectives and adverbs (*Cancho, Solé & Köhler, 2004*). Although more or less convoluted syntactic structures can be encoded in sentences, the simplest form of the syntactical dependencies is a subject being specified as an object. For instance, in the sentence "love is weakness", "love" is specified as "weakness" through the verb "is". The verb "is" does not encapsulate any intrinsic meaning but it can be replaced by a link between "love" and "weakness". Syntactic dependencies can be more general, for instance in the sentence "the cat sat on the chair" the nominal subject "cat" is linked to the object "chair" through the verb "to sit", which retains some meaning. The determiner "the" and the preposition "on" do not retain meaning by themselves but connect the other parts of speech and are thus essential for extracting syntactic dependencies. The specification of such dependencies was implemented through the *TextStructure[]* command in Mathematica 11, which produces all syntactic relationships between the parts of speech of a given sentence. From the resulting network of directed dependencies, prepositions and auxiliary verbs were removed as nodes and replaced by syntactic links. In order to avoid the inflection of lemmas with the same meaning, i.e., having "weak" and "weakness" in the same network, word stemming was performed at the network level.

After the extraction of syntactic dependencies, the resulting syntactic network identified a set of connected, stemmed concepts. Syntactic structure can be informative about conceptual links in the mental lexicon and be predictive of language learning and processing (*Cancho, Solé & Köhler, 2004*; *Stella et al., 2018*). However, syntactic dependencies neglect the possibility of using and exchanging synonyms in the same syntactic structure providing the same meaning (e.g., saying "he has a quiet character" conveys the same meaning of saying "he has a quiet nature"). In order to account also for meaning overlap, the syntactic structure was enriched with synonym relationships

from WordNet 3.0 (*Miller, 1998*), as implemented in the curated repository *WordData[]* available in Mathematica 11. Meaning overlap between concepts is also representative of semantic memory patterns in the mental lexicon, with previous works showing how synonym networks can predict language learning and facilitate conceptual navigation (*Stella et al., 2018*; *Siew et al., 2019*). Notice that in this analysis, "word" and "concept" are used interchangeably.

After the construction of the network layers of syntactic relationships and of synonyms, every concept was connected either by syntactic or semantic links. From the affective mega-study of *Warriner, Kuperman & Brysbaert (2013)*, each concept was endowed with a valence label, e.g., "positive", "neutral" or "negative". These labels represented the average valence attributed to each concept by a population of individuals involved in a large-scale behavioural study rating more than 13,900 English words (*Warriner, Kuperman & Brysbaert, 2013*). Words were classified as positive, neutral or negative, respectively, according to their location in the upper quartile (higher valence), interquartile range (neutral valence), or lower quartile (lower valence) in the distribution of valence scores for over 9,000 different word stems. These valence scores were obtained from averaging over words with the same stemmed root.

A single TFMN was made by considering all 10k tweets together, with syntactic links bridging words within the same sentence. Since the same words could be repeated across sentences, the overall syntactic network bridged different sentences together, either from the same or from different tweets.

Figure 1A provides an example of a text-based forma mentis network. A TFMN can be represented either as an edge-coloured graph or as a multiplex network (*Stella et al., 2018*). In network science, multiplex networks contain different sets of relationships/associations between elements of the same group, e.g., the same set of words being linked either through synonyms or syntactic dependencies. All the links expressing a certain type of relationship form a so-called network layer. A multiplex network is then a collection of multiple network layers, each one featuring replicas of the same set of nodes, e.g., see Fig. 1A. In edge-coloured graphs, replicas are aggregated together and all different relationships are expressed on one layer, e.g., see Fig. 1A. In TFMNs, where there is no explicit transition between syntactic and semantic replicas of the same word/node, these two representations are equivalent and their only purpose is to distinguish between syntactic and semantic links.

Figure 1A provides a pictorial representation of the way a TFMN is built from text. The starting point is the assumption that knowledge and emotions about the real world are encoded in people's mental lexica. Individuals express their opinions, combining knowledge, sentiment and emotions, through language. TFMNs are based on the syntactic/semantic links between words as extracted from text. The resulting network structure identifies the organisation of conceptual associations as expressed by online users in social discourse. Network neighbourhoods correspond to the syntactic/semantic *frame* of concepts associated to a given word and investigated by the so called *semantic frame theory* (*Fillmore, 2006*). Analysing the associates of a concept gives information about: (i) how it was framed in terms of associated meanings, (ii) how it was perceived in terms

of valence and emotions. With reference to Fig. 1B, a concept linked to mostly positive concepts acquires by association a positive connotation in text, which corresponds to a positive *aura* on the TFMN. In the following, a *valence aura* will identify the most frequent sentiment label (e.g., positive, negative, neutral) attributed to the semantic associates of a concept, analogously to previous studies (*Stella et al., 2019*; *Stella, 2020*). Also *emotional auras* will be considered, indicating the contextual feelings and emotions elicited by the conceptual associates of a word in a given semantic frame.

## Additional cognitive data

In order to test the power of text-based forma mentis network in identifying relevant concepts in text, additional cognitive data was used as benchmark. Semantic similarity and relevance of scientific concepts, as extrapolated from short scientific texts, was investigated and compared against free association data from the Complex Forma Mentis project (*Stella et al., 2019*). The benchmark text adopted here was the booklet "Complexity Explained", co-authored by several researchers in complexity science and composed of 7 short paragraphs describing one specific concept each (cf. https://complexityexplained.github.io/, Last Accessed: 16/03/2020). The analysed paragraphs were about: "interactions", "emergence", "dynamics", "self-organisation", "adaptation", "interdisciplinarity" and "methods". The results of such benchmark are reported at the beginning of the Results section.

## Emotional profiling and emotional flowers

Compared to previous approaches using behavioural forma mentis networks (*Stella, 2020*; *Stella & Zaytseva, 2020*; *Stella et al., 2019*), the current analysis introduces also an emotional profiling of stances as encapsulated in the reconstructed mindsets. Emotional profiling is defined in terms of considering how many associates of a concept, in the TFMN, elicit a given emotion. Concepts linked to a negation (e.g., "not") were transformed into their antonyms, so that both original concepts and their negated meanings were considered for reconstructing an emotional profile. Considered emotions included:

- **Anger**, a negative emotion representing reactions of irritation and rage towards an external threat;
- **Disgust**, a negative emotion indicating aversion and closure;
- **Fear**, a negative emotion indicating a need to actively avoid and prevent potential threats;
- **Trust**, a positive emotion indicating openness towards the outer world;
- **Joy**, a positive emotion of excitement and satisfaction;
- **Sadness**, a neutral emotion, neither positive nor negative, indicating states of sorrow, thoughtfulness and inhibition;
- **Surprise**, a neutral emotion relative to being upset or startled by an unexpected event;
- **Anticipation**, a neutral emotion indicating one's projection into future events, including desire or anxiety.

The above constitute basic building blocks of a wide spectrum of emotional states (*Ekman & Davidson, 1994*). For a more detailed description of emotions, please see *Ekman & Davidson (1994)* and *Mohammad & Turney (2013)*. The mega-study by Mohammad

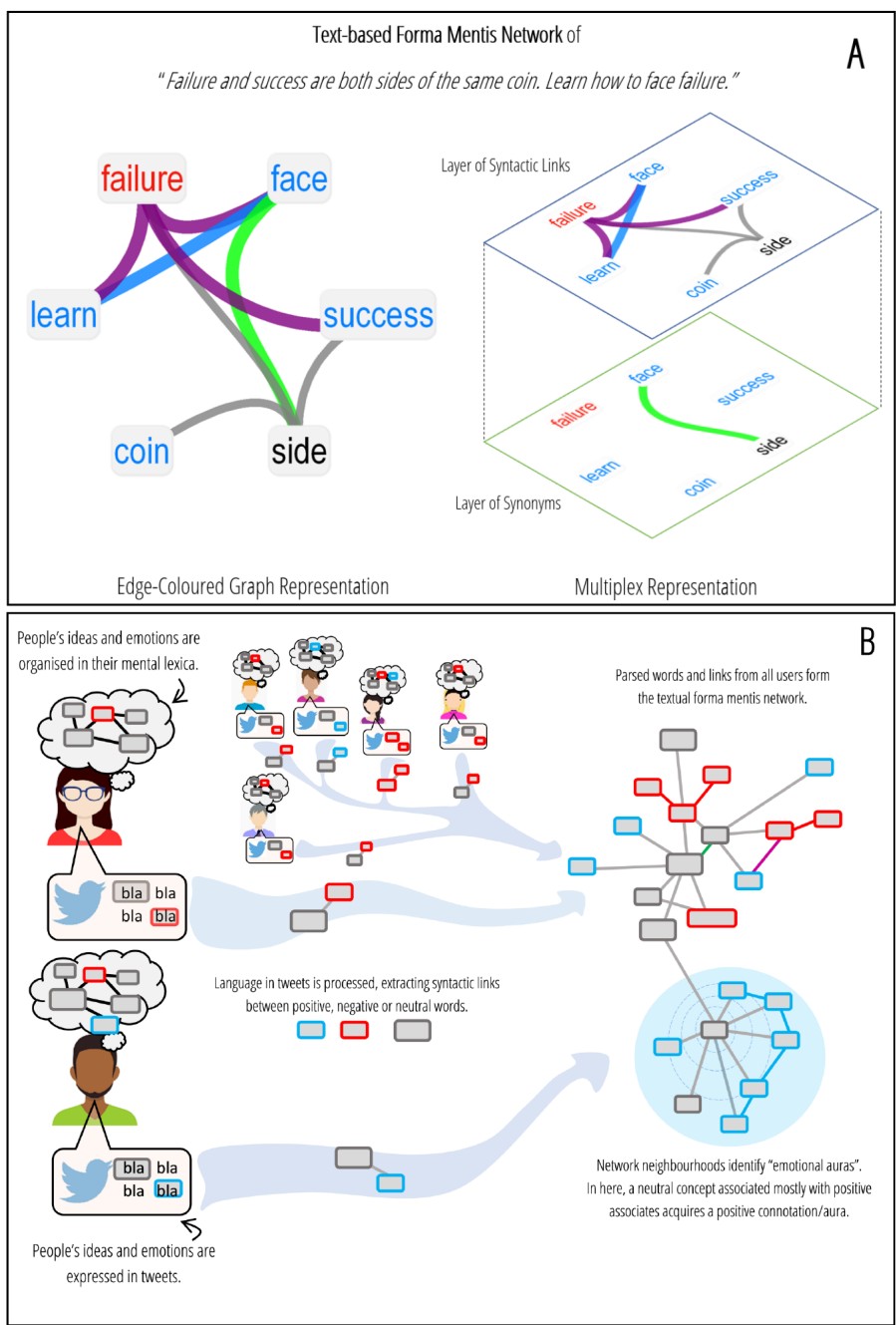

**Figure 1** (A) Example of a text-based forma mentis network. A TFMN can be represented either as an edge-coloured graph or as a multiplex network. Positive (negative) words are highlighted in cyan (red). Neutral words are in black. Syntactic links between positive (negative) words are highlighted in cyan (red) too. Syntactic links between positive and negative concepts are in purple. All semantic links of meaning overlap are highlighted in green. (B) Infographics about how a TFMN is assembled. Individuals organise their knowledge and emotional perception of the real world in their mental lexicon (comic clouds). People express their mental stances on Twitter through text messages. Language from text is processed and syntactic/semantic links between words are aggregated in order to form a TFMN. Network neighbourhoods define the emotional auras of concepts.

and colleagues is also known as the Word-Emotion Association Lexicon, indicating how human participants associated individual concepts to emotional states. In other words, the resulting dataset indicated which emotions were elicited or *evoked* in human subjects reading individual concepts. The study used crowd-sourcing on Mechanical Turk for achieving a large-scale mapping of English words (including 14,000 unique lemmas) and was used also in other successful investigations about affect patterns (cf. *Mohammad & Turney, 2013*).

Comparing the emotional richness of different semantic frames is problematic, as those frames might involve different numbers of words. Comparing an observed emotional profile with a null model represents a more appropriate comparison, especially considering that there are more words eliciting fear and joy than anticipation and sadness in the dataset. A statistical comparison can answer questions like "Was trust the most prominent emotion just because there are more words eliciting trust in language?".

As a null model for comparison with an observed emotional profile, random sampling was used here. Fixing the same number $m$ of words eliciting at least one emotion and observed in an empirical semantic frame, $m$ random samples were selected uniformly at random from the Word-Emotion Lexicon. The emotional profile of this random sampling was then computed and stored. This operation was repeated 1,000 times, obtaining a random distribution of emotional profiles from which mean, standard deviation and z-scores were computed. The expected mean and standard deviations were used in visualising the emotional richness of a semantic frame through a bar plot. Instead, z-scores were organised in a bar sector chart reminiscent of the atlas of emotions (*Ekman & Davidson, 1994*). In visualisation, the rejection region relative to $z < 1.96$ is a semi-transparent circle. Bars are shaped like petals, one per emotion, and coloured differently. Those petals going out of the rejection region indicate a semantic frame with jargon eliciting a given emotion more than random expectation (within a significance level of 0.05). Concentric rings indicate units of z-scores after 2. For their resemblance with flowers, these visualisations are called *emotional flowers*.

## Cognitive measures of semantic similarity on language networks

This work built networks of conceptual associations between concepts. Several independent studies have reported that on such networks, metrics like network distance are powerful proxies for quantifying semantic relatedness (*Kenett et al., 2017*; *Stella & Zaytseva, 2020*). Network distance $d_{ij}$ is defined as the minimum number of links (here conceptual associations) connecting any two words $i$ and $j$ on a network structure (*Siew et al., 2019*). On multiplex lexical networks, links of any colour/layer can be used (*Stella et al., 2018*; *Stella, 2019*). Measures based on network distance such as closeness centrality have been found to identify also concepts of relevance from a cognitive perspective when detecting key words for word learning (*Stella et al., 2018*; *Stella, 2019*), language processing (*Kenett et al., 2017*; *Siew et al., 2019*; *Castro & Stella, 2019*) and knowledge exploration (*Stella & Zaytseva, 2020*; *Akimushkin, Amancio & Oliveira Jr, 2017*; *Amancio, 2015*). Closeness centrality (*Siew et al., 2019*) $c(i)$ is attributed to node $i$ by checking how distant it is to its connected

neighbours, in formulas:

$$c(i) = \frac{N}{\sum_{j=1}^{N} d_{ij}}. \tag{1}$$

Notice that the above formula applies to fully connected components of a network but does not enable direct comparison of components including different numbers of nodes (e.g., a component including only 5 nodes versus a component with 1,000 nodes). Remember that TFMNs are multiplex but do not feature any weight or inter-layer links, hence for them multiplex closeness is equivalent to the single-layer closeness on the aggregate network obtained by considering all edge types as equivalent, as considered in previous cognitive approaches (*Stella et al., 2018*). Therefore, multiplex closeness on the TFMN considers both syntactic and semantic links when outlining shortest paths between concepts.

Forma mentis networks included also structural features of the mental lexicon of individuals combined with affective patterns (*Warriner, Kuperman & Brysbaert, 2013*). This unique combination allowed for *Stella et al. (2019)* to introduce the network metric of *valence auras*, i.e., the mode of valence labels in a given neighbourhood. In *Stella et al. (2019)* a word was defined as having a positive (negative) valence aura if mostly linked to positive (negative) words. This measure was used also in the current analysis, although it has to be underlined that in here these affect measures represented how a large-scale population, independent from the one producing the texts, perceived individual concepts. Since valence auras still depended on the connectivity of conceptual associations as assembled by text authors, this metric quantified how groups of these authors structured and perceived their knowledge. In the current analysis, valence auras rather than single-word valence labels were key to reconstructing positive/negative perceptions of individual concepts by checking their conceptual associates in the analysed online discourse.

## RESULTS

The main results of this manuscript are twofold. On the one hand, the current analysis provides a cross-validation of the information revealed by TFMNs through a benchmark on short texts, revolving around specific topics. Comparison with validated behavioural cognitive data indicates that the structure of TFMNs captures semantically relevant information about text like topics or conceptual relevance. On the other hand, the rest of the Results section focuses around reconstructing and analysing a data-driven picture of the online perception of women in science and the gender gap as reported by online users on Twitter.

### Benchmark of text-based forma mentis networks on short texts

This subsection reports results of the benchmarking analysis of "Complexity Explained" (see Methods) through TFMNs. For each of the seven paragraphs of the booklet a TFMN was built. Closeness centrality was used for identifying the most central concepts in every TFMN and produce rankings of the most relevant words in each text. Overall, the resulting

**Table 1  Top-ranked concepts in the TFMNs obtained from the seven paragraphs of the booklet *Complexity Explained*.** The ranking is based on closeness centrality. Every paragraph revolves around one topic, clearly established by the authors and reported here too.

| Rank/Topic | Interactions | Emergence | Dynamics | Self-organisation | Adaptation | Interdisciplinarity | Methods |
|---|---|---|---|---|---|---|---|
| 1 | component | system | system | pattern | adapt | system | compute |
| 2 | interact | property | change | emerge | system | understand | model |
| 3 | whole | part | state | may | able | science | method |
| 4 | study | component | behaviour | organ | become | complex | mathematics |
| 5 | system | whole | point | become | function | use | lead |
| 6 | make | sum | show | interact | damage | variety | require |
| 7 | difficult | phenomenon | variable | system | evolve | manage | involve |
| 8 | part | complex | dynamic | produce | go | domain | analysis |
| 9 | new | exhibit | tend | property | may | ecology | forecast |
| 10 | consist | deduce | depend | lead | robust | biology | rule |

forma mentis network contained a median of 49 concepts and were fully connected. Table 1 reports the 10 most central words in each network and their underlying topic.

Table 1 indicates that, beyond an overall agreement between key concepts and topics, as identified by TFMNs, the network topology of syntactic/semantic associations identified the most distinctive conceptual features of topics. For instance, while the ranking for the topic "Interactions" reported mainly concepts relative to structure (e.g., *component, part, whole, consist*), the ranking of "Dynamics" identified concepts related to the evolution over time of a system (e.g., *change, state, behaviour, dynamic*). Words expressing resilience and robustness to attacks, like "damage", "function", "evolve", "adapt" and "robust", were found to be central in the textual forma mentis network of "Adaptation". TFMNs detected "domain" as being a relevant concept in the "Interdisciplinarity" paragraph. This was expected in a text describing complexity science as an umbrella for different research areas. The "Methods" paragraph revolved, as expected, around quantitative concepts, e.g., *forecast, compute, model, method, analysis, rule*. In the same paragraph, mathematics was found to be highly relevant, in agreement with the overall necessity of a mathematical language for investigating complex systems (*Cancho, Solé & Köhler, 2004*; *Siew et al., 2019*).

Although the above qualitative analysis indicated an overall agreement between topics and identified key concepts, a more quantitative approach was further pursued. With the aim of assessing whether the identified concepts were more or less semantically related to the topics designed by text authors, free associations and semantic network distance were adopted. In networks of free associations, nodes/words are linked if they elicit a quick recall of each other in a behavioural task (*De Deyne et al., 2019*). Previous studies have shown how semantic network distance on networks of free associations are a good proxy of semantic relatedness, superior also to semantic latent analysis (*Kenett et al., 2017*). Although several datasets for free associations are available in the literature, this benchmark considered two: the large-scale Small World of Words gathered by *De Deyne et al. (2019)* and the small-scale Complex Forma Mentis project gathered by *Stella et al. (2019)*. Many of the scientific terms present in Complexity Explained were absent in the Small World of

Words but present in the STEM free associations provided by *Stella et al. (2019)*. Therefore, this analysis focused on the second dataset of free associations.

The network distance between every relevant word and its reference topic was computed for all paragraphs within the the network of free associations by *Stella et al. (2019)*, i.e., a behavioural forma mentis network based on the collective mindset around science of 59 complexity researchers. Hence, relevant words were extracted from textual forma mentis networks (of syntactic/semantic relationships). The semantic proximity of these relevant words with their reference topic was tested on a behavioural forma mentis network (of free associations). The reference topics were "interaction", "emergence", "dynamics", "self-organisation", "interdisciplinary" and "methods" and the relevant concepts were the ones in Table 1. The resulting set of empirical semantic network distances was then compared against a reference null model with randomised TFMNs having the same number of links and nodes of the original networks but with randomly reshuffled links (i.e., configuration models *Stella et al., 2018*). The reshuffling disrupted semantic relationships between network structure and meaning (*Stella et al., 2018*). A location test between the empirical and the randomised network distances (over 50 network realisations) indicated a statistically significant difference between the clustering of relevant words around each topic and the null model (Mann–Whitney test, $U = 17827$, $p$-value: $0.0267 < 0.05$) at a significance level of 0.05. In the networked free associations representing the scientific knowledge of complexity researchers, the words identified as relevant for a topic by TFMNs tended to be closer to their own topic (median network distance: 3.1) than on the randomised networks (median network distance: 3.7). Since semantic network distance on networks of free associations indicates semantic relatedness (*Kenett et al., 2017*), these results confirmed that closeness centrality on TFMNs was capable of identifying concepts relevant to the specific topic underlying a given text.

Given the successful outcome of such benchmark, TFMNs can therefore be applied to extracting relevant features of other short texts. The following section reports on the results of TFMNs when used for analysing over 10,000 tweets about gender gap in science.

## Analysis of the perception of gender gap in STEM with cognitive networks

The textual forma mentis network obtained from processing the selected tweets included a largest connected component including 3,005 stemmed concepts and 28,004 connections (24,693 in the syntactic layer and 3,311 in the synonyms layer). The 10 most central concepts in terms of closeness centrality were: "STEM", "science", "we", "do", "you", "learn", "make", "get", "work", "need". Notice that closeness centrality captures semantic prominence, i.e., a concept being semantically related to all other concepts connected to it. The identification of "STEM" and "science" as semantically prominent in the dataset confirms the scientific scope of the latter. This finding importantly indicates that TFMNs are able to identify semantically prominent concepts not only in well-ordered, short texts (see the benchmark) but also in the collection of short texts represented by tweets. This should not be given for granted, since tweets also include hashtags, emojis and linguistic abbreviations that do not appear in well-ordered paragraphs and which were not included

in TFMNs. Despite this limitation, TFMNs identified words capturing the general context of science of the investigated online discourse. Concepts like "woman" and "man" ranked 103rd and 246th, respectively, mirroring the relevance of women in science for the selected online discourse. "Scientist" ranked higher, at the 54th position, further confirming the scientific scope of the dataset.

### Conceptual associations with and around a concept identify word clusters

The TFMN was more clustered than random expectation (using a single-layer clustering coefficient where all links are aggregated together, cf. *Siew et al., 2019*). Concepts linked to a common neighbour tended to get connected with each other too. The empirical network displayed a mean clustering coefficient of 0.327 ($0.166 \pm 0.007$ for reference configuration models *Stella et al., 2018*). Hence, in the TFMN concepts clustering around a given word (e.g., "STEM") shared syntactic/semantic links too, a tendency that can provide a richer structure about the conceptual organisation of knowledge around specific concepts/topics in terms of (more) conceptual links. Consequently, investigating clustered networked neighbourhoods of words can provide contextual information that would be lost by considering either words in isolation or only the list of associates to a given concept. For this reason, the investigation focused on word clusters in order to better understand the online perception of the STEM gender gap.

### Valence auras and global network metrics highlight an overall positive online stance towards STEM and gender gap

The 3,005 stemmed concepts extracted from relevant tweets and connected in the forma mentis network were rated as positive (1045), negative (430) and neutral (1943) according to the procedure described in the Methods. Positive concepts were found to have a higher median degree than negative concepts (Mann–Whitney test, numerosity $n_1 = 1045$, median degree $k_1 = 9$, $n_2 = 430$, $k_2 = 4$, $U = 3 \cdot 10^6$, $p < 10^{-6}$). Positive concepts were also found to be more central in the reconstructed mindset in terms of requiring fewer syntactic/synonymy associations in order to reach any other connected concept, i.e., centrality as expressed by multiplex closeness (*Stella et al., 2018*), (Mann–Whitney test, numerosity $n_1 = 1045$, median degree $k_1 = 0.3601$, $n_2 = 430$, $k_2 = 0.3307$, $U = 3 \cdot 10^6$, $p < 10^{-6}$). These comparisons indicate that in the analysed corpus of twitter language focusing on women and STEM, positive concepts were more predominant, more well connected and more central than negative concepts. The richer network structure of conceptual associations of positive concepts translated into a generally positive attitude of twitter users towards the STEM gender gap.

Figure 2A reports the emotional auras (*Stella et al., 2019*; *Stella, 2020*) of hub concepts in the reconstructed mindset of social media users. While single-word labels were determined from the sentiment of a general population, i.e., participants in an affect mega-study (*Warriner, Kuperman & Brysbaert, 2013*), auras emerged from the specific syntactic/synonymy relationships traced in the analysed data and therefore characterised the reconstructed mindset of the population of interest. Concepts like "STEM", "gender" and "gap" are neutral in commonly spoken language but were associated mostly with positive concepts in the language of the online discourse, i.e., surrounded by a positive

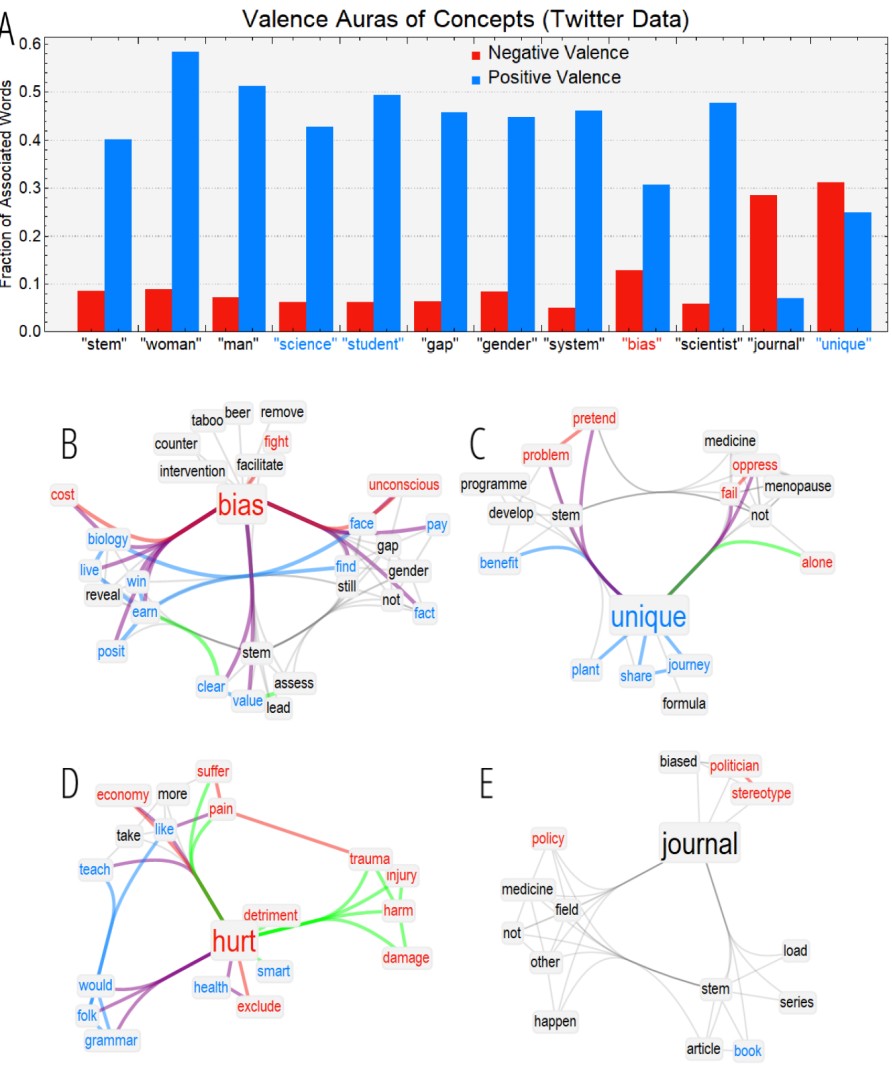

**Figure 2** (A) Valence auras in the global forma mentis network of all 10,384 analysed tweets. Positive (negative) words are highlighted in cyan (red). A fraction of 0.4 indicates that 40% of the neighbours of a word (e.g., STEM) are labelled as positive. (B–E) Textual forma mentis network for words linked to the target topics like "bias" (B), "unique" (C), "hurt" (D) and "journal" (E).

emotional aura. This indicates an overall positive perception of these topics that would go *undetected* when investigating "STEM", "gender" and "gap" in isolation. Differently from other investigations with forma mentis networks (*Stella et al., 2019*; *Stella, 2020*), the sampled online audience reported a strongly positive perception/aura of "student". Also "bias", a negative concept, was surrounded by a positive emotional aura, indicating an overall mixed stance trying to figure out positive aspects of a bias (e.g., how to overcome it). Further analysis is required in order to better understand the above perceptions. Figure 2 reports the mindset structure around "bias" (B) and other concepts like "unique" (C), "hurt" (D), and "journal" (E). The forma mentis neighbourhood of "bias" included associations to positive concepts like "face", "win", "clear" and to negative concepts

like "fight", "cost" and "stereotype", all revolving around contrasting and overcoming biases. Therefore, the reconstructed TFMN indicates how social media users discussed about "bias" in terms of a negative entity to be contrasted, overcome and won, thus explaining the above mixed perception. Other words associated to bias gravitated around the gender pay gap and economic implications (*Courey & Heywood, 2018*), e.g., "earn", "cost", "value". Links with "unconscious" and "fact" indicate that social media users were aware of hidden, unconscious gender biases affecting the gender gap (*Pietri et al., 2018*).

Figure 2C also reports how "unique" was associated in the online discourse around the gender gap. The mixed perception of "unique" included negative associations to concepts eliciting loneliness, e.g., "oppress", "fail", "alone" and "pretend", which are unexpected when considering the positive perception of "unique" itself. Hence, the TFMN indicates that when associated in messages focusing on women in STEM, the meaning of "unique" shifted from positive to mixed and included negative connotations of social exclusion and sense of failure. This is an example of *contextual valence shifting* (*Polanyi & Zaenen, 2006*), a phenomenon in which a concept changes its valence according to its semantic context. TFMNs represent a valuable quantitative tool for identifying valence shifting through semantic associates and emotional profiling.

The multiplex structure of TFMNs can influence an emotional aura. Synonymy links (green) do not come from text but rather from a pre-determined lexicon of synonyms (i.e., WordNet *Miller, 1998*) that has general validity over common language. Instead, syntactic (blue/red/gray) associations come from the specifically analysed language. In this way, TFMNs can identify also missing conceptual associations or the predominance of synonymy over syntactic links.

The negative aura of "hurt" in Fig. 2D was mostly due to general synonyms whereas specific syntactic associations included positive concepts like "teach", "like" or "health", thus indicating a lack of hurtful/painful conceptual associations in the investigated online discourse over women in STEM. Network structure was informative also about the negative aura attributed to "journal", which was linked to "politician", "stereotype", "bias" and "policy". These syntactic links indicated a social awareness about journals and news media potentially perpetrating gender stereotypes, a role that was investigated in previous studies (*Steinke, 2017*).

## The online social perception of "woman", "man" and "person" reconstructed by TFMN

Although it is expected for the online discourse to feature also automatic accounts and social bots, previous research identified human users as being more predominant in driving and diffusing message exchanges on social platforms (*Stella, Ferrara & De Domenico, 2018*). Comparing how people identify themselves in a discourse where they are the main drivers can be informative about social roles and users' self-perception (*Varol & Uluturk, 2020*).

Figure 3 reports and compares the forma mentis network around "woman" (A), "man" (B) and "person" (D). All these three concepts, commonly perceived as positive entities, were surrounded by positive emotional auras in agreement with the overall positive features reported above of the whole TFMN.

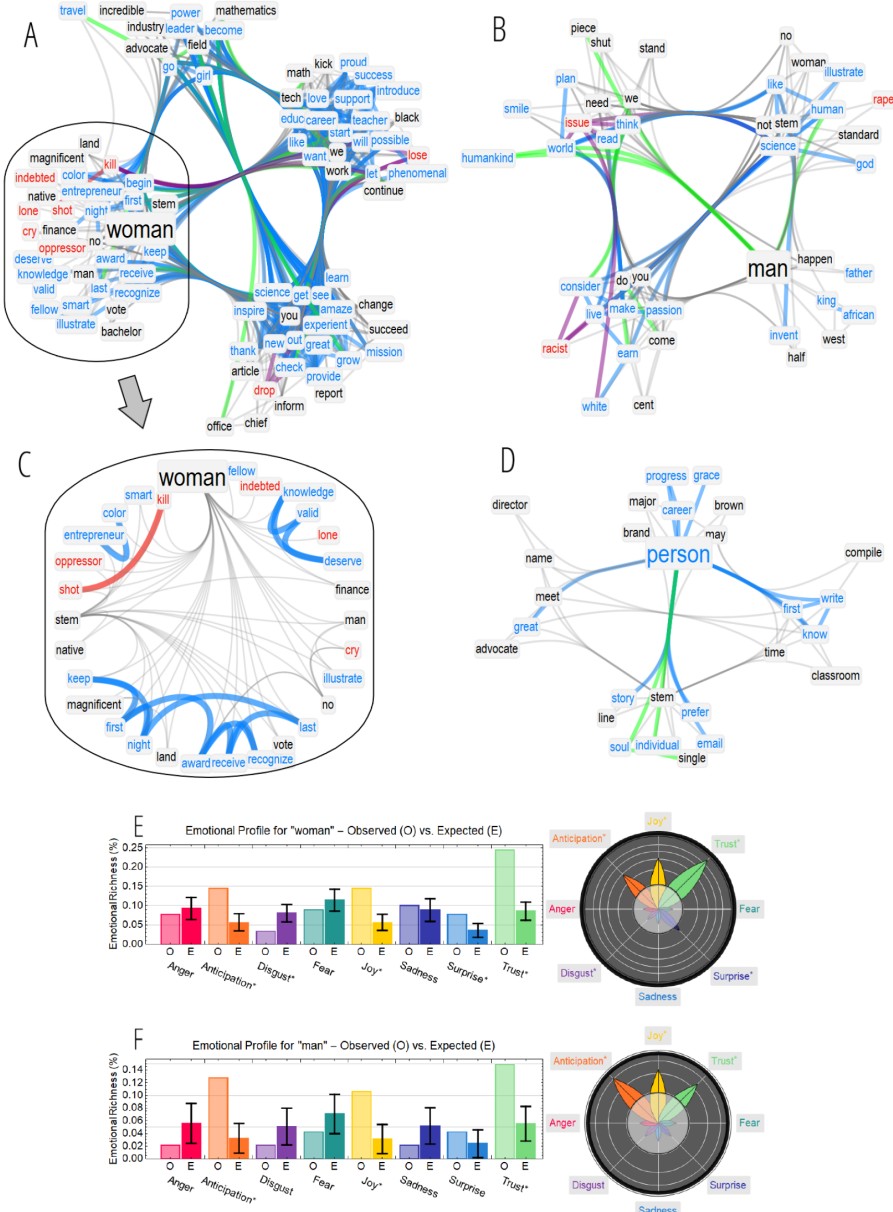

**Figure 3** (A–D) TFMN frames around "woman" (A), "man" (B) and "person" (D). Words are clustered in communities obtained via the Louvain algorithm (cf. *Konstantinidis, Papadopoulos & Kompatsiaris, 2017*). Words in the same community of "woman" in A are plotted through a hierarchical edge-bundling visualisation in C. Positive (negative) words and links are highlighted in cyan (red). Links between positive and negative concepts are reported in purple. Semantic links between synonyms are in green. (E–F) Emotional profiles for the frames of "woman" (E) and "man" (F) indicate the fraction of concepts in a frame eliciting a certain emotion. Comparison against random sampling is reported, with z-scores plotted as emotional flowers (see Methods).

The reconstructed social perception of "woman" within the online discourse of gender gap in STEM identified mainly semantic clusters expressing three aspects:

1.  leadership and professional recognition in STEM (e.g., "enterpreneur", "award", "recognize", "leader", "power", "career", "success", "deserve", "science", "valid");
2.  learning and education (e.g., "teacher", "inspire", "learn");
3.  social welfare (e.g., "finance", "advocate").

All these aspects were dominated by positive, concrete concepts eliciting a sense of achievement and professional establishment. The resulting picture was an overall positive sentiment/perception of the online social discourse about the figures of women in science. However, within this positive landscape, a closer look at the local community of concepts clustering around "woman" identified also negative associations, further highlighted in Fig. 3C.

The analysed text associated women also to "indebted", "kill", "lone" and "cry", highlighting a multifaceted perception of women in STEM and including negative traits that might originate from *stereotype threat* (*Shapiro & Williams, 2012*; *Pennington et al., 2016*). In cognitive psychology, stereotypes can advantageously form a belief about or characterise key traits of groups of people through little cognitive effort. Stereotypical perceptions come at the cost of being not grounded in empirical knowledge, providing inexpensive but potentially erroneous information that can: deeply affect perception and cognitive processing (*Pennington et al., 2016*), induce anxiety, and reduce performance under pressure, e.g., causing a sense of threatening. For instance, STEM stereotypes like "girls are not good at maths" have been reported to affect even the self-perception of female STEM students, causing unconscious biases which reduced girls' performance and retaining of STEM subjects (*Shapiro & Williams, 2012*; *Steinke, 2017*; *Chavatzia, 2017*). The TFMN reported in Fig. 3A, and its inset (C), highlighted the stereotype of the "oppressive, lonely, big-shot woman in STEM", relative to women in STEM achieving successful careers only at the cost of sacrificing empathy and other positive personality traits (*Ely, Ibarra & Kolb, 2011*; *Madsen & Andrade, 2018*). Recent studies have overwhelmingly exposed such stereotype, cf. *Steinke (2017)*, and the permanence of these negative associations in online social discourse represents evidence of how such stereotype is still deep-rooted in daily communication. Notice that this might have also positive repercussions, as improving the awareness of the effects of stereotype threat by simply talking about it can boost self-perception and promote the implementation of effective countermeasures fighting stereotypes (*Madsen & Andrade, 2018*).

The reconstructed online perception of women was overwhelmingly positive and it indicated how women's success and leadership in STEM deserve recognition (*Madsen & Andrade, 2018*). This perception has to be compared against the one of males. Figure 3B considers the forma mentis network around "man", which included mostly positive concepts like "science", "passion" and "smile". However, syntactic links associated "man" also to negative concepts such as "rape", "issue" and "racist", indicating the presence in the online discourse of forms of accusations or condemnations of the role played by men in social issues like sexual harassment, rape and racism (*Karami et al., 2020*). In the forma mentis around "man", special attention has to be devoted also to the negative

particle "not", which is syntactically linked to "god", "think", "make" and "consider". The negation of all these concepts, as indicated by the syntactic links, provided a multi-faceted perception where men in STEM were related to a superiority complex, a stereotypical self-perception of superiority in science-related achievements, for instance in mathematics assessments (*Leyva, 2017*), promoted by preliminary and incomplete studies (cf. *Leyva, 2017*; *Chavatzia, 2017*). This quantitative result, embedded in language and highlighted by forma mentis networks, indicates that fighting the gender gap in STEM means also changing men's stereotypical roles in science.

In terms of emotional profiles, both "woman" and "man" included associates evoking mainly trust, joy and anticipation, see Figs. 3E–3F. The TFMN around "woman" was slightly richer in anger-eliciting words, like "kill" or "shot", than the TFMN around "man", but both levels were compatible with random expectation.

In Fig. 3D, the TFMN around "person" was devoid of any negative concept or sign of stereotype threat. Word clusters related "person" to a multifaceted perception about career progression, mindfulness and education. The associates of "person" evoked mostly trust (emotional profile not shown for brevity), further confirming the positive perception of "person" itself. Given the gender-less dimension of "person" in English, it is interesting to underline how the above negative perceptions are more tightly connected to gender-rich words for human beings like "man" and "woman" and not to genderless words like "person". Since words in language do not only describe reality but rather contribute to forging it (*Malt, Wolff & Wolff, 2010*), the above result suggests that reducing the incidence of the gender gap might be possible also by speaking more and more in terms of "persons in STEM" rather than in terms of contrasting "men versus women in STEM", always while respecting individual differences.

## The online social perception of gender gap and scientist stereotypes as reconstructed by TFMN

This subsection investigates "gender" and "gap" and their online perceptions. Figure 4 considers the TFMN around "gender" (A,B) and "gap" (C,D), which were both considered as neutral concepts in language.

The online discourse over the social platform associated these two concepts with each other, as expected, and related both of them mainly to positive words. Hence, the overall online emotional aura of "gender gap" was mostly positive but it also included some negative associates. The community structure for the TFMN of "gender" identified three perceptual dimensions in the way online users talked about gender:

- a positive dimension of respect and consideration (with associates like "balance", "close", "consider" and "respect");
- a research dimension in understanding the science of gender (with associates like "research", "support", "future", "highlight" and "explain");
- a mostly negative dimension of gender imbalance and biases (with associates like "doubt", "imbalance" and "attack").

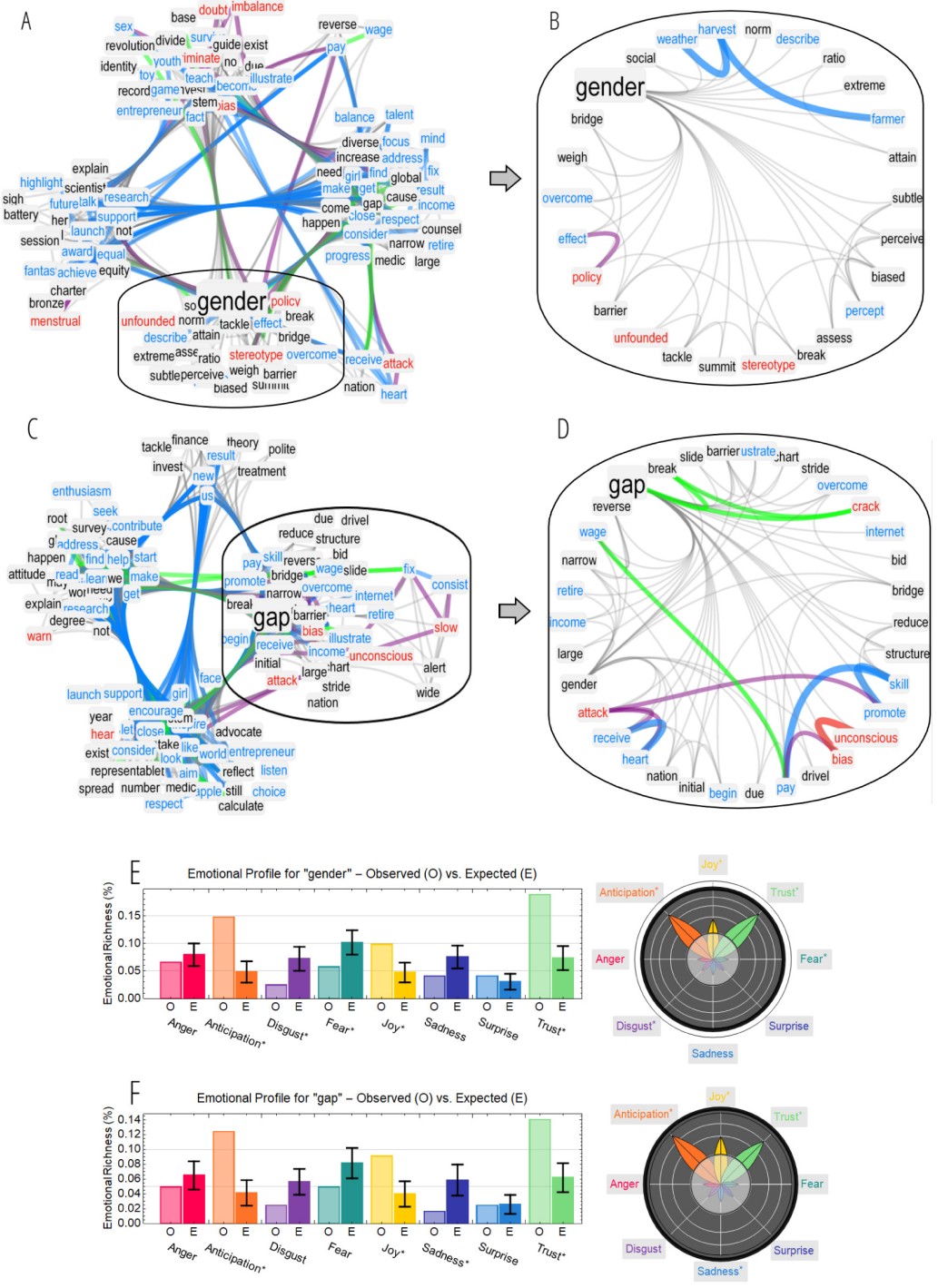

**Figure 4** **(A–D) TFMN frames around "gender" (A) and "gap" (C). Words are clustered in communities obtained via the Louvain algorithm (cf. *Konstantinidis, Papadopoulos & Kompatsiaris, 2017*).** Words in the same community of "gender" in A and "gap" in C are plotted through a hierarchical edge-bundling visualisation in B and D, respectively. Positive (negative) (continued on next page...)

**Figure 4 (…continued)**
words and links are highlighted in cyan (red). Links between positive and negative concepts are reported in purple. Semantic links between synonyms are in green. (E–F) Emotional profiles for the frames of "gender" (E) and "gap" (F) indicate the fraction of concepts in a frame eliciting a certain emotion. Comparison against random sampling is reported, with z-scores plotted as emotional flowers (see Methods).

Associations of "gender" with concepts like "stereotype", "unfounded", "break" and "tackle" (see Fig. 4B) indicated an attitude of opposition, among online users, against gender stereotypes in the considered online communications.

Analogously to "gender", also "gap" included negative associations to "bias" and "attack" and positive links to concepts like "overcome", "close" and "consider" (see Fig. 4C). The conceptual cluster around "gap" highlighted in Fig. 4D contained associations mostly focusing on:

1. the expectation to overcome and reduce the gender gap in the future (e.g.,"reverse", "break", "crack", "bridge" and "reduce");
2. the economic impact of the gender gap (e.g., "pay", "income", "wage").

These two dimensions indicate a positive intention for the online discourse to tackle and reduce gender gaps, while displaying awareness about the gender pay gap (*Courey & Heywood, 2018*) and also an emotion of anticipation or projection of such challenge in the future (see emotional profiling in Figs. 4E–4F). This provides additional quantitative indication for the importance of achieving equal wages in STEM in order to reduce the gender gap, in agreement with previous studies (*Courey & Heywood, 2018*). Figure 4 contains also negative associations between "gap", "bias" and "unconscious", indicating a remarkable awareness of online users about the gender gap being rooted in unconscious bias and gender preconceptions that are difficult to detect and act upon (*Karami et al., 2020*; *Steinke, 2017*; *Moss-Racusin et al., 2012*; *Pietri et al., 2018*).

As already mentioned above, a prominent mechanism of unconscious bias is stereotype threat, where unconscious perceptions cause anxiety and negative latent emotions that influence performance, e.g., girls aware of the preconception that "girls are not good in science" end up performing worse than males in STEM tasks (*Moss-Racusin et al., 2012*; *Shapiro & Williams, 2012*; *Madsen & Andrade, 2018*; *Chavatzia, 2017*). Given the awareness of online users about unconscious biases, it becomes relevant to explore the networked TFMN mindset in search of signs of stereotype threat.

Because of the STEM scope of the dataset, the focus here was devoted to detecting stereotypical perceptions in the figure of "scientist", whose neighbourhood is reported in Fig. 5A. The TFMN quantified positive semantic associations surrounding "scientist". Clusters of more tightly associations identified a strong perception of scientists in relation to innovation (e.g., "tech", "society", "entrepreneur", "innovate", "discover", "leadership"). A closer look (cf. Figure 5B) revealed also enthusiasm-eliciting concepts (e.g., "great", "notable", "famous", "celebrant", "dream", "congratulate") and career-related jargon (e.g., "showcase", "assess", "work", "peer", "academia"). Hence, scientists were perceived as successful professionals by online users, analogously to what other social groups like

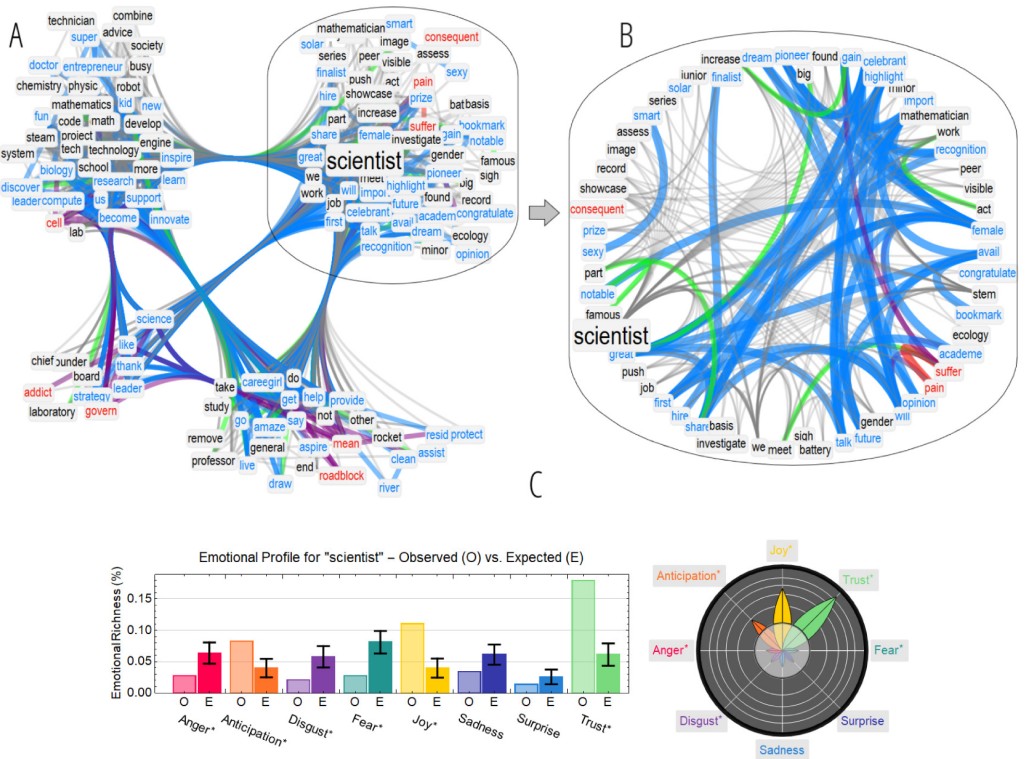

**Figure 5** (A–B) Semantic frame for "scientist" (A). The plot on the right zooms in the community of words more tightly connected to "scientist" (B). Positive (negative) words and links are highlighted in cyan (red). Links between positive and negative concepts are reported in purple. Synonymy relationships are in green. (C) Emotional profile and flower for the neighbourhood of "scientist", indicating what emotions elicit the associates of "scientist" itself in the analysed corpus.

high-school students did in other studies (cfr. *Stella, 2020*). Even within this success-centred perception, online users linked scientists to concepts like "suffer", "pain" and "gain", displaying awareness about the cost of success and the need for hard work.

Overall, the mental construct of "scientist" represented by the TFMN identified a well-rounded, balanced and positive online perception of scientists in terms of successful, hard-working professionals, devoid of any significant patterns of stereotype threat associating scientists only to a male-gender sphere, differently from stereotypical perceptions found in other datasets and groups via the Implicit Association Test (*Lane, Goh & Driver-Linn, 2012*; *Steinke, 2017*).

In line with the semantic analysis, according to the emotional profile in Fig. 5C, scientists elicited strong feelings of trust, joy and anticipation.

### A frequency analysis confirms the above general positive perception

The above approach relies on the syntactic structure of language used in tweets, which is indicative of the way online users discussed the gender gap. However the results relying on language structure should be cross-validated with other approaches like frequency

counts, which neglect linguistic structure but keep into account word repetitions in a given discourse.

In the considered dataset, after stemming words, their frequency of appearance across tweets was counted. 430 negative words were featured in the considered dataset and appeared in total 1,192 times. The most frequent negative stemmed words included "problem", "lack", "miss", "critic", "pain", "doubt", "old", "kill", "forget" and "toxic". Some of these concepts were featured in the semantic frame of "woman", as reported above. The dataset included also 1045 positive words, which appeared in total 13,905 times. The most frequent positive stemmed words included "girl", "career", "student", "learn", "new", "support", "great", "like", "research" and "thank". Most of these words were featured in the semantic frames of "woman", "man", "person" and "scientist". 1943 neutral words appeared in total 16,711 times. Hence, the considered social discourse represented as a TFMN here not only featured more than twice as many positive words than negative concepts but positive words also occurred with a total frequency more than ten times higher than the one of negative words. These patterns confirm the results obtained with the TFMN about the considered social discourse around the gender gap being mostly dominated by positive concepts but also modestly riddled with negative concepts. In comparison with this simplistic frequency approach, the network structure investigated up above provided insightful results on the ways positive and negative words were used and linked across the above investigated semantic frames of "woman", "man", "scientist", "gender" and "gap".

## DISCUSSION

As outlined in recent Big Data Analytics studies (*Huang et al., 2020*; *Odic & Wojcik, 2019*; *Chavatzia, 2017*), differences in gender discrimination are strongly influenced by distorted mindsets, with deep, often hidden, repercussions over pay gaps (*Courey & Heywood, 2018*) and sexual harassment (*Karami et al., 2020*). Combining these findings with the ever-increasing influence of social media over real life (*Jansen et al., 2009*; *Waqas et al., 2019*; *Stella, Ferrara & De Domenico, 2018*; *Nasar, Jaffry & Malik, 2019*) highlights an urgent necessity for using information processing in order to understand "if" and "how" specific massive online social platforms promote information on distorted mindsets. The tackling of such research question is essential for countering gender biases with data-informed approaches (*Huang et al., 2020*). This work tackled this question by extracting, reconstructing and understanding with textual forma mentis networks (TFMNs) how online users perceived and discussed the topics of "women in STEM" and the "gender gap" on Twitter.

As a knowledge extraction technique (*Nasar, Jaffry & Malik, 2019*), TFMNs indicated an overall positive attitude of the online discourse towards tackling and reducing gender inequality. Going beyond the positive/negative polarity of standard sentiment analysis (*Mohammad, 2016*; *Stella, Ferrara & De Domenico, 2018*), this work attributed TFMNs with a richer emotional profile, using emotional data from (*Mohammad & Turney, 2013*) and providing a quantitative way for detecting *contextual connotation shifting* (*Polanyi &*

*Zaenen, 2006*), i.e., a concept being perceived in different ways according to its semantic context and associates. The reconstructed forma mentis networks outlined "man", "woman" and "scientist" as being associated with concepts eliciting predominantly trust, joy and anticipation and, with way less intensity, also anger and fear. Trust is a feeling of confidence, security and positive endorsement (*Ekman & Davidson, 1994*), its predominance in the retrieved TFMNs suggests a willingness for online users to provide and exchange endorsements to each others' messages while debating the topic of "women in science". This result is in contrast with a previous study reporting Twitter as being more prone to host general-level negative rather than positive emotional content (*Waterloo et al., 2018*). This disparity might be explained by considering that Twitter interactions focus mainly over weak social ties, including mostly acquaintances and casual contacts rather than strongly personal relationships (*Waterloo et al., 2018*). Since the expression of topic-inspired negative emotions along weak social ties is considered being less acceptable (*Ferrara & Yang, 2015*), more positive emotional profiles would be expected from topic-specific Twitter public debate, and not from general level content sharing like the one in (*Waterloo et al., 2018*). As a future research direction, it would be interesting to detect whether the mostly trustful and positive perception of gender biases would persist also on other social media platforms with stronger social ties such as Facebook or WhatsApp (*Waterloo et al., 2018*).

Another emotional state prominently featured in all the reconstructed stances towards "woman", "gender", "gap", "man" and "scientist" was anticipation, a neutral emotional state including either pleasure or anxiety towards future events (*Ekman & Davidson, 1994*). The predominance of anticipation and joy with respect to other emotions like sadness, fear, disgust or anger, suggests the prevalence of positive expectations in the analysed text, examples being the excitement-related concepts associated with "scientist" or the successful dimension attributed to women in STEM. These patterns indicate the occurrence of messages celebrating women in science and their success, contrasting the gender gap with stories of excitement and professional achievement. Extensive research (cfr. *Pietri et al., 2018*; *Chavatzia, 2017*; *Madsen & Andrade, 2018*) indicates that promoting professional achievements of women in science has strong beneficial effects in favouring women's representation in STEM, as it enables girls in identifying relatable and inspiring stories of success in STEM going beyond discrimination.

The reconstructed TFMNs reported evidence for online users being aware about unconscious gender biases (*Shapiro & Williams, 2012*). These biases occur when an individual consciously rejects gender stereotypes but is still influenced by and makes unconscious evaluations based on such stereotypes themselves, see also (*Madsen & Andrade, 2018*; *Ely, Ibarra & Kolb, 2011*). At the individual level, unconscious and passive discrimination based on gender stereotypes can have smaller repercussions in comparison to actively promoting gender inequality. However, at the group level, many unconscious small biases can interact in a complex systems fashion (*Hogue & Lord, 2007*) and lead to the emergence of "powerful yet often invisible barriers" of gender discrimination (*Ely, Ibarra & Kolb, 2011*). These barriers undermine considerably women representation and professional growth in a variety of fields including also STEM (*Shapiro & Williams,*

*2012*; *Lane, Goh & Driver-Linn, 2012*; *Madsen & Andrade, 2018*; *Huang et al., 2020*; *Odic & Wojcik, 2019*). Forma mentis networks captured signals of unconscious gender bias in online discourse around "women in science". These conceptual links can be beneficial in promoting awareness about the above invisible barriers, further suggesting the extreme importance of fighting implicit biases in STEM careers for closing the gender gap in STEM (*Pietri et al., 2018*).

Another prominent topic emerging from the TFMNs is the gender pay gap, a mismatch between the salaries of individuals of different genders performing the same job (*Courey & Heywood, 2018*). The online perception reconstructed by forma mentis networks indicates that pay gaps are closely semantically related to both "gender" and "gap", thus indicating that closing the gender pay gap is key for fighting gender biases in STEM, in agreement with previous relevant studies (*Ely, Ibarra & Kolb, 2011*; *Courey & Heywood, 2018*).

Although partial evidence for a stereotypical, angry-eliciting and fearful perception of women in STEM as "lone survivors" was present in the neighbourhood of "woman", the overall perception of such concept was positive and elicited mostly trustful and joyous concepts, celebrating women's success in STEM. The reconstructed role of "scientist" did not include stereotypical conceptual associations, indicating a lack of stereotype threat phenomena (*Shapiro & Williams, 2012*) in the considered online discourse. In cognitive psychology, stereotype threats represent unconscious mechanisms that affect performance of a given group in relation to the stereotypical expectations commonly shared about that group (e.g., "girls are bad at maths" is a stereotype harming girls' performances in maths, cfr. *Pennington et al., 2016*; *Chavatzia, 2017*). Forma mentis networks mostly related the online perception of scientists to success and career progression and highlighted a lack of stereotypical perceptions, differently from the detection of scientist-centred stereotypes found in previous approaches with cognitive network science (*Stella, 2020*), via the Implicit Association Test (*Lane, Goh & Driver-Linn, 2012*). Such virtuous finding might be the consequence of the relatively high participation of STEM professionals over the Twittersphere, which can disrupt stereotypical perceptions. As future research, it would be interesting to apply TFMNs for detecting potential patterns of stereotype threat in other social platforms like Facebook or WhatsApp, that are more exposed to negative content (*Waterloo et al., 2018*) and also less prone to hosting posts from STEM experts. It would also be of interest to investigate different patterns of social denounce about gender-related issues like workplace sexism or discrimination towards LGBT+ people.

From a methodological perspective, TFMNs rely on the recent theory of cognitive networks (*Siew et al., 2019*) and include syntactic and semantic conceptual associations which are informative of the structure of knowledge perceived by text authors (*Stella et al., 2018*; *Stella, 2019*). Differently from other successful models of knowledge representation such as concept maps (*Koponen & Nousiainen, 2019*; *Dóczi, 2019*) and knowledge graphs (*Amancio, 2015*; *Akimushkin, Amancio & Oliveira Jr, 2017*; *De Arruda et al., 2019*), forma mentis networks contain also emotional information, outlining sentiment and emotional patterns in the way individuals assembled their stance in a text. This contextual information is essential for the interpretability of a detected stance (*Nasar, Jaffry & Malik, 2019*). The combination of rational knowledge and emotional perceptions is the main strength of

this methodology. Combining knowledge and emotions is particularly relevant when investigating problems as complex as the gender gap, which spans several cognitive, educational and organisational aspects (*Easterly & Ricard, 2011*; *Beede et al., 2011*).

The framework of textual forma mentis networks reported here has some important limitations. Differently from the behavioural forma mentis networks introduced in previous studies (*Stella & Zaytseva, 2020*; *Stella et al., 2019*; *Stella, 2020*), in TFMNs the valence of concepts represents population-level averages extracted from mega-studies in psycholinguistics (*Warriner, Kuperman & Brysbaert, 2013*). This representation assumes an overall shared perception of the emotional content of concepts that might be preserved at the global level, i.e., on an online platform where large numbers of users of multiple backgrounds interact with each other. However, this assumption might be violated within specific populations. For instance, *Stella et al. (2019)* showed that a population made entirely of high-school students perceived "maths" as a negative concept whereas a population of international researchers perceived the same concept as positive. The overall perception of "maths" reported in the language norms by *Warriner, Kuperman & Brysbaert (2013)* was neutral, as reported also in the TFMNs presented here (*Warriner, Kuperman & Brysbaert, 2013*). It is important to keep this assumption in mind when applying TFMNs to mindset reconstruction in specific and non-heterogenous populations. Nonetheless, even in these populations TFMNs can be informative about conceptual associations as expressed by the semantic/syntactic multiplex network structure, since the emotional attitude towards a concept can be reconstructed from the attitude and meaning of its associates (*Polanyi & Zaenen, 2006*). As a future research direction, a potential approach for achieving population-specific valence labels would be valence extraction from text, a technique that exploits word embedding and machine learning (*Mohammad, 2016*; *Rudkowsky et al., 2018*) and as such works well for longer texts like books or essays but is still relatively less accurate for shorter texts like tweets or posts, which are less rich in semantic information (*Polanyi & Zaenen, 2006*). Another limitation of this study was neglecting emojis and hashtags included in tweets. These elements were not included in the exploratory analysis here because they do not possess clear syntactic dependencies like words do. Future research could translate emojis, hashtags and even abbreviations in full words, connect them in TFMNs through relationships of semantic overlap, and test the impact that their addition could have in terms of enriching the stance expressed in online social discourse. Despite neglecting emojis and hashtags, TFMNs were still able to identify concepts of semantic prominence in the considered dataset, mostly because the tweets gathered here were mostly at the beginning of conversations and were thus richer in common language than hashtags or emojis. In these tweets, by construction, emojis and hashtags played a more peripheral role in expressing meaning.

Another assumption underlying TFMNs is treating modifiers of meaning (e.g., negations) as nodes equivalent to concepts. Modifiers can alter the meaning expressed by individual concepts, providing a contextual richness that is not captured by the so-called "bag of words" models (*Polanyi & Zaenen, 2006*; *Rudkowsky et al., 2018*), where a text is represented by an unstructured list of its concepts. Textual forma mentis networks provide syntactic and semantic contextual background to concepts so that the investigation of

modifiers cannot be independent on the analysis of conceptual associates. For instance, in the current approach, negations were considered in the emotional profiling of conceptual neighbourhoods in the following way: antonyms of words linked to negations were added to emotional counts, providing information about opposite meanings in addition to the concepts originally available in the TFMN (e.g., if "appreciation" was connected to "not" then its antonym "disgust" was considered in the emotional profiling too). Other ways of grading the intensity of statements or negations could be tested and implemented within future work.

It has to be underlined that the dataset investigated here did not include users' conversations, e.g., replies without the considered keywords. This choice was motivated by the framing of TFMNs, which combine linguistic knowledge and emotional perceptions as directly coming from people's mindsets/mental lexica (see Fig. 1B). This requires an immediate translation of thoughts into texts without being excessively influenced by other users' behaviour on the online social platform. Message production in conversations is considerably more dependent on other users' behaviour than conversation starting (*Tagg, 2015*). Reading the stance promoted by the beginning of a conversation can give rise to a wide variety of behaviours like agreeing with an influential user just because of their fame or trolling a user because of their background, includinng frequently observed extremes like flaming, i.e., starting insulting discussion without concrete logical arguments. Trolling, flaming and social bias can greatly alter social discourse and the perception of a given topic because of dynamics taking place within the online social platform but not encoded in the mental lexica of online users. For instance, a lack of social cues and exposure to other users' replies were found to boost the likelihood of online trolling or flaming (*Tagg, 2015*). The mostly positive perceptions reported in this investigation indicate that the starting points of discussion for the gender gap in Twitter report a mostly trustful perception of the gap, modestly riddled with traces of gender stereotype threat. Assessing whether this mostly trustful initial perception evolves through conversation posting, trolling and flaming represents a fascinating direction for future research. It also indicates that Twitter policies aimed at reducing discriminating, abusive language should employ strategies flagging more thoroughly users' replies and central parts of conversations.

Another future research direction would be the implementation of TFMNs in synergy with social network analysis, much alike what recent approaches suggested (*Rodrigues & Pietrocola, 2020*; *Stella, Ferrara & De Domenico, 2018*), in order to better capture online social behaviour (*Varol & Uluturk, 2020*), expose the exact semantic frame used by texts for presenting ideas or educational material (*Koponen & Nousiainen, 2019*), highlight distorted attitudes towards science, education and educational systems (*Stella et al., 2019*; *Stella, 2020*) or better understand misinformation spread (*Pierri, Artoni & Ceri, 2020*).

## CONCLUSIONS

Textual forma mentis networks (TFMNs) provide contextual conceptual and emotional information from text, providing a rich picture of how text authors perceived and associated multiple topics. When applied to the social media discussion of women in science, TFMNs

identified a mostly positive, trustful and anticipation-rich discourse but also highlighted online awareness about relevant issues like unconscious gender biases and the gender pay gap. The application of TFMNs to the analysis of online discourses opens new ways for quantitatively assessing the role played by social media in promoting/hampering gender biases and distorted mindsets. Textual forma mentis networks can open new ways of accessing social media, understanding the content of online communication and providing new bridges for linking social dynamics with cognitive and emotional information spread.

## ACKNOWLEDGEMENTS

The author acknowledges Prof. Jim Jansen for valuable feedback on this work.

### Funding

The author received no funding for this work.

### Competing Interests

Massimo Stella is employed at Complex Science Consulting, which had no influence over this study. Consequently, the author declares he has no competing interests.

### Author Contributions

- Massimo Stella conceived and designed the experiments, performed the experiments, analyzed the data, performed the computation work, prepared figures and/or tables, authored or reviewed drafts of the paper, and approved the final draft.

### Data Availability

The tweet IDs are available as Supplementary Material.

### Supplemental Information

Supplemental information for this article can be found online at http://dx.doi.org/10.7717/peerj-cs.295#supplemental-information.

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
