# Peer review of "Text-mining forma mentis networks reconstruct public perception of the STEM gender gap in social media"

_PeerJ Computer Science, doi:10.7717/peerj-cs.295_

## Round 0.1 · original submission · Minor Revisions

All three reports have basically pointed out some that minor issues should be addressed. Most importantly, it is interesting to address the comments regarding some methodological clarifications. Most of the required clarifications are mentioned by reviewer #3. Overall, the paper is well written and no additional experiments are required at this point.

Reviewer 1 ·

Basic reporting

The paper treats a relevant and debated topic in a structured and methodologically sound manner. Overall I found it very easy to follow.

I have very few minor suggestions:
1) The caption of Figure 1 should explain what black words and grey links are. They are likely to be associated with neutral sentiment but it is better to be clear.
2) The caption of Figure 2 has inverted top/bottom labels.
3) At page 10 the subsections have unneeded numbering

Experimental design

No comment

Validity of the findings

No comment

·

Basic reporting

The manuscript “Text-mining forma mentis networks reconstruct public perception of the STEM gender gap in social media ” introduces the applications of Forma Mentis Network (FMN)-approach to explore how gender gap in STEM (science, technology and mathematics education) is perceived in social media. The research reported in the manuscript continues the research line the author has developed in previous publications, mentioned in refs. Stella et al 2019,Stella & Zaytseva 2019, Stella 2020. In the new contribution, the method of analysis to explore the mental mind sets through textual analysis is developed further, and it is called Textual Forma Mentis Network (TFMN). The new method makes use of edge-coloured graphs and the multiplex structure of the networks. The FMN methods as it was applied to lexical networks has gradually, step-by-step, been develop to current TFMN method which is methodologically quite advanced and powerful, and is developed in a way that it can adapted to big data sources. I find it exciting and welcome new technique that opens up entirely new types of approaches in science education research and subjects related to STEM. These kinds of methods provide valuable complementary information to more traditional studies based on phenomenographic or interpretative studies, which are common in case of gender gap and gender issues.

The topic of gender gap is issues related to gender bias are extensively researched topics in science education research and no common single reasons for the existence of gap and bias are found. However, it is clear that expectations, stereotypic images and role models play key roles in formation of these gaps. The present study approached the problem from viewpoint how public perceives the gender gap in discussing STEM related topics. This is well chosen viewpoint, which provides background information what kinds of related views, beliefs and conception the public discussion reveals. It is still far from being able to resolve the issues, but provides a big picture of how the topic is perceived. The author anchors the study and its viewpoint very carefully in the recent research, which demonstrates that biased and distorted mind-sets are closely connected to the gender gap. The TFMN method is then designed and adapted to reveal indicators of such distorted minds sets from the twitter messages.

The context of the study and the current understanding of how gender gap and mind-sets related to gender issues in STEM are carefully discussed in the manuscript and a very good and insightful summary is provided. As such, the summary is very valuable because it not only summarise the recent research but also provides a balanced synthesis of what is known of the topic. It also provides strong argument in favour to focus on analysing views as they are expressed in public discussion. Such contribution shows a good scholarship and good acquaintance with the subject and is valuable contribution on the current research.

The method (TFMN) is then very fluently and even seamlessly developed to answer the questions raised. The study and its approach is thus skilfully composed so that the questions it attempts to answer the theoretical basis and the applied methods form a coherently composed whole. The study is exemplary in that regard.
The exposition of the material, results and discussion are all very clearly done. The language and structure of the manuscript appear to me quite perfect (though I am not native in English language) throughout. In addition, the stucturisation of the text makes it easy to follow and the argumentation folds out easily and with clarity.

Experimental design

The experimental design is for analysis of datasets from social media. The dataset is here the online perception of the gender gap in STEM as retrieved from Twitter. The collection and curation of the dataset is explained and documented carefully and in details. The method to construct the textual forma mentis network is explained in main text, and in more details in the supplementary material. In addition, the data includes the cognitive data used for detecting conceptual overlap in meaning between words and their valence. The data sets are rich and they support each other, providing basis for very thorough picture of the phenomena, which are studied.

The method of analysis is developed on basis of work already reported in Stella et al. 2019 and Stella 2020. The new TFMN method is also tested carefully used the previous forma mentis networks. In addition, in the construction of networks the synonymous expression are paid careful considerations and taken into account. These aspects greatly increases the reliability of the method. Method is also described in detailed enough way to allow replication of the study in similar contexts and by using similar data collection.

The only aspect, the author could consider is if the meaning of edge-coloured graphs and multiplex networks needs to be explained in a bit more accessible way after or before figure 1. Probably these concepts are well known for most readers of PeerJ, but in the lucky case that also science education researchers find this paper, it might be helpful to provide a sentence or two to explain those concepts in more detail. However, I leave this as an option to be considered by author.

Validity of the findings

The context of application of the method is to my knowledge new. The context is very important, currently extensively researched but still in need of better understanding of the factors affecting the formation of gender gap and biased attitudes. The present work bring a new viewpoint on this problem, and provides new powerful tools to analyse the problem. The results of the study are discussed from many perspectives. In addition, the limitations of the study are pointed out. What I find as a particular strength of the study is how it manages to join the emotional and conceptual aspects, as they are present in the twitter discussions. This shows the complexity of problems, where conceptual and rational aspects are mixed with emotional ones.

The results are nearly self-evidently valid within the scope of the study and how it is framed; the public view of gender gap as it appears on twitter. Of course, the twitter tweets and social media discussions reveal what topics are of interest, how public picks out topics they tweet etc., and as such provide window to the mind-sets. This is part of the problem, but then, in addition, there are factors related to institutional organisation, social dynamics and societal factors which affect how the mind-sets expressed in e.g. twitter are formed. Of course, it is not reasonable to assume that in the present study, such dimensions need to be discussed, but in the end of the study, authors note that it could be interesting to extend the study in direction of social dynamics and sociosemantics. I think this is very good idea and the methods the author presents, seems to be perfectly well adapted for such extensions.

Additional comments

The author have produced a very interesting study, which introduces powerful methods to explore mind-sets and provide information on basis of big-data. The topic of the study, the gender gap, is very important and currently one of the most researched topics in STEM, but still many questions remain open. The present study is a valuable new view on that discussion. Methodologically, the study demonstrates the power of network approach in complex issues related to STEM research and helps to see new, complementary ways to advance research and provide access to information not available easily or even at all by using methods that are more traditional. I think, that research in STEM urgently needs to adapt this kinds of new approaches and start making use of the network science and its methods to make progress, and even more importantly, to start to conceptualise the problems from complex system perspective. I hope that this study will find it ways to hands of scholars and researcher in science education, since I believe that it would have a significant impact in that area of research.

Reviewer 3 ·

Basic reporting

I thought this paper had an interesting combination of methods applied to an important topic. However, I also found the story hard to follow in several places, and I ended with several questions about the analysis and its implications. So there's a lot of questions/suggestions below, but I hope they will help to clarify and strengthen the paper.

Language: The article was written in technically correct English and was unambiguous at the sentence level. There were some ambiguities/unclear areas at the conceptual/paragraph level, which I comment on below.

Literature/background: Appropriate prior literature was referenced. I believe some additional background is needed to help link the ideas required to understand this paper. There are several complex elements in the "story"--text analysis theory, implicit bias and the gender gap in STEM, and network analysis--and very few readers will have expertise in all areas. Figure 1 does a good job of clarifying what makes up a TFMN. It’s less clear how that network object connects in the analysis to "valence auras." Similarly, "mental lexicon" is referred to frequently in the introduction, but I'm not sure what it actually means in the analysis. Is it possible to add a schematic (possibly expanding Figure 1), showing either a "toy example" of the rest of the analysis steps, or a conceptual diagram of how the stages of analysis connect to each other? I found the paper very interesting but also very hard to follow, because the theory is so elaborate, and I kept flipping back and forth to try to understand how all the pieces fit together.

Structure, figures, tables, raw data: The article format follows standard sections, and the figures are relevant. Much of the figure text is small and difficult to read without magnification, especially on the network diagrams. There were several other clarity issues for figures:

Questions about figures
1. Figure 3: I've read this caption several times and still don't follow, especially the "Top" part. "Man" is in the top row, not the bottom row as is stated. The bottom/top and left/right structure described in the caption doesn't seem to match the structure of what’s shown. For example, the bottom left position does not seem to show all neighbors of the word "women" (as the caption says), but an extracted subset. The top/bottom distinction for the "Networks on the right" is also confusing. Does the bottom right panel show all words that were connected to "person," which happens to be many fewer than the number connected to "man," of which only a subset are shown here?

Basically, I think the attempt to explain this figure using rows and columns doesn't help because there are too many qualifiers and exceptions to the categories even within the four-panel "Top" part of this figure. (In contrast, this structure worked much better with Figure 4.) Talking about the "top of the top" and "bottom of the top" just muddies things further. I suggest breaking Figure 3 into at least two figures (the "Top" and "Bottom" parts), and at least consider further breaking apart the "Top" to more clearly step through what each panel of it is showing. Having two views of the “woman” network makes it very hard to follow which one is analogous to what’s happening in the right-hand column (confusingly, I think the top-left view is analogous to the bottom-right panel, and the bottom-left is showing the community view, analogous to the top-right?).

2. On Figures 3, 4, and 5, the emotional profile plots have different vertical scales, which makes it hard to compare them and is visually misleading (e.g., at a glance the profile for “women” has fewer joyful associations than “men,” but it actually has a higher percentage). If it’s meaningful to compare those plots between Figures 3, 4, and 5, they should be on the same scale--if it’s not meaningful, that should be explained.

Minor issues:
Figure 2 caption: The "Top" and "Bottom" labels are switched.
Figure 4 caption: The "Bottom" caption should say "gender" and "gap" rather than "woman" and "man," I believe.

The full raw data is not available, but Tweet IDs to retrieve it are provided.

Self-contained, relevant results: I believe the paper is self-contained, after some clarifications.

Experimental design

Aims and scope: The paper is within the stated aims and scope of the journal.

Research questions and identified knowledge gap: The paper does not state a specific research question, but explores one application of TFMNs. The knowledge gap around this application is well articulated.

Rigorous investigation/Methods described in sufficient detail: Several parts of the data collection and analysis need to be clarified. My data collection questions also bear on the authors' claims in the Results and Discussion (further comments on that in the Validity and Findings section of this review).

Data collection questions:
1. Were all tweets in the thread gathered, or just the ones containing the four search terms? This is important because many substantive conversations on Twitter happen around multi-tweet threads, and some tweets may lack the keywords even though they are on the same topic. I can see arguments for or against including all threaded tweets, but the paper should clarify which approach was used.
2. Similarly, were comments (replies) from other users collected and analyzed, or only the original tweet (plus any comments that also had keywords)? Excluding comments from the analysis could seriously skew the findings, because it omits the insults, harassment, and general "trolling" that often come in response to online discussions of gender gaps in STEM. If comments were not included, this should be stated clearly, along with the rationale.
3. Less a question than a comment, but the set of keywords used to find tweets seems like it might filter for a pretty academic set of conversations. ("Gender gap," in particular, is the kind of phrase I hear mostly in diversity talks, so seems likely to find people already "in the system.") This is relevant to the gender gap in STEM, of course. But there are other conversations that might be less likely to show up--for example, people talking about workplace sexism they experience, or the gendered slurs and other pushback that come up when people talk about this online (especially women or gender non-conforming folks).

Analysis questions:
1. "Concepts" and "words" seem to be used interchangeably. If that's true, it should be stated explicitly. Otherwise, places like page 9, line 389 or page 10, line 451-452, are confusing because saying "concepts" implies an additional layer of analysis (beyond words -> nodes, find most central nodes).
2. A single TFMN was made by pooling all 10K Tweets together, right? Are syntactic network links only drawn within sentences, or can they bridge sentences in a Tweet? If they can cross sentences, how are borders between Tweets in a thread handled?
3. These networks are multiplex (with syntactic and semantic edges), but the closeness measure defined is for single-relation networks. How was it adapted for multiplex networks (alternately, how were the two edge types combined for use in equation (1))?
4. Are the networks weighted by frequency of occurrence, or unweighted? If they’re weighted, how is that used in closeness centrality calculations? If they’re unweighted, isn’t that throwing a way a lot of information?
5. Page 10, line 419, refers to “the networked mindset of 59 complexity science researchers.” I'm not sure what this refers to--were the "Complexity explained" paragraphs by 59 different people, or what is this?
6. The benchmark text from the Complexity Explained booklet doesn't seem to be very similar to the target text (thousands of tweets). For that reason, it's not clear to me that it's a good benchmark. Does performance on a few well-ordered paragraphs mean that we can assume that the method works for tweets, which are filled with hashtags, abbreviations, informal phrasing, and emoji? It would be good to at least comment on the differences, and why this benchmark is believed to be appropriate anyway.

Validity of the findings

Underlying data: The raw data is not provided, but it should be possible to reconstruct the data set given the list of tweet IDs. Source code is provided for analyzing a cleaned data set.

Conclusions: As stated in part 2, I'm concerned that the results may have been biased toward positive words/content if only tweets and not their replies were used. Or it's possible that positive voices in the replies would "prevail"--but if they weren't analyzed, that isn't known, and should be clearly stated as a limitation. The abstract says that "This analysis thus identified Twitter discourse as promoting a mostly stereotype-free, positive/trustful perception of gender disparity, of relevance for closing the gap." This is a fairly sharp contrast to the many complaints about Twitter's failure to impose meaningful anti-harassment policies. It's possible that the subset of words used to gather tweets filtered out most of that phenomenon. Alternately, maybe there really is more positive-aspected talk, and people just remember the harassment and threats more because of the damage they cause. Regardless, I think this study will come off as disconnected from reality if it makes that claim in the abstract without also discussing Twitter's famous problems in this regard, and how the author interprets the results in light of those issues.

Additional comments

To reiterate, I think this is a neat combination of methods + problem, but in its current form I think the paper has several serious ambiguities. I hope my comments are useful.

---

## Round 0.2 · accepted · Accept

Reviewer #1 considers that all issues have been addressed. Reviewer #3 did not accept to re-review this paper. I have checked his/her comments and my analysis confirmed that all other issues have also been addressed.

Reviewer 1 ·

Basic reporting

The author resolved all my concerns about the manuscript.

Experimental design

no comment

Validity of the findings

no comment

Additional comments

no comment